# The Protection of the Historic City: The Case of the Surroundings of the Lonja de la Seda in Valencia (Spain), UNESCO World Heritage

Camilla Mileto 🔟 and Fernando Vegas López-Manzanares *🔟

Centro de Investigación en Arquitectura, Patrimonio y Gestión para el Desarrollo Sostenible (PEGASO), Universitat Politècnica de València, 46022 Valencia, Spain; cami2@cpa.upv.es
* Correspondence: fvegas@cpa.upv.es

**Abstract:** In geographical terms, historic cities possess an inertia in regard to the modification of urban function. This explains why buildings may change over time, but the location of the functions remains. For over a thousand years, the city of Valencia has concentrated the commercial activity of its historic centre around the building of the Lonja de la Seda, its surrounding buildings, and its adjacent spaces, streets and squares. Recent constructions coexist with centuries-old buildings, witnesses to the transformations of this urban enclave, which has retained its commercial function. Although the Lonja de la Seda was declared World Heritage by UNESCO in 1996, its surroundings, despite being of interest and closely linked to the protected building, were not. This article analyses the history and evolution of the built fabric and urban spaces of this complex, which represents the nerve centre for commerce in the city of Valencia. This text presents research based on studies carried out directly on the buildings in this context by the authors, as well as indirect examinations of documentation from the archives and the existing bibliography. The aim of this study is to showcase how combining material and documentary studies can lead to a broader definition of the tangible and intangible values of cultural heritage. This, in turn, could lead to the comprehensive enhancement of the historic city, where historic residential fabric and notable buildings are merely manifestations of the process for the construction of the city.

**Keywords:** historic city; city centre; protection; Valencia; historic dwellings; UNESCO World Heritage; buffer zone; typology



## 1. Introduction, Theoretical Context, Scope and Methodology

The Lonja de la Seda (originally, the Silk Exchange) in Valencia, built between 1482 and 1548, was declared World Heritage by UNESCO in 1996 under the criteria "i: to represent a masterpiece of human creative genius" and "iv: to be an outstanding example of a type of building, architectural or technological ensemble or landscape which illustrates (a) significant stage(s) in human history". These values highlighted by UNESCO in the declaration are clearly evident. However, would this highly spectacular building exist without its surroundings? Or rather, would the Lonja de la Seda make sense without its surroundings? (Figure 1).

### 1.1. Some Definitions for Further Exploration

The concept of surroundings has progressively been defined throughout the theoretical reflection of architectural heritage. Since the 18th century, interest in history and architecture has gradually encouraged the concept of monument, derived from the Latin term "monumentum" (commemorative monument), which is in turn derived from the verb "monere", "to make you think of something" or "to remind somebody of something". However, over time, this concept linked to a single unique building such as the Lonja, has

progressively expanded to cover more open and inclusive concepts and to define assets of cultural interest.

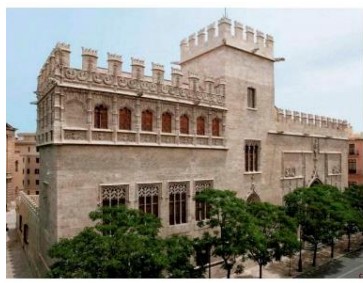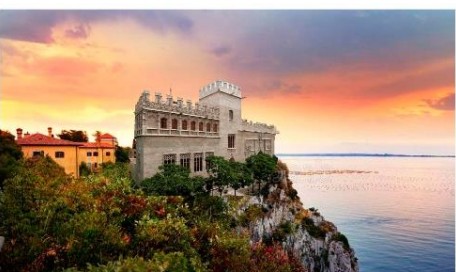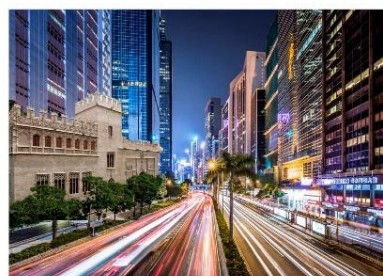

**Figure 1.** Three images of the Lonja de la Seda in three different contexts: its current context; the simulation of an environment on the edge of the sea and the simulation of a new urban city environment (source: authors).

The Spanish Law for Historical Heritage states that "within the Spanish Historical Heritage, and for the purpose of granting greater protection and safeguarding, the category of asset of cultural interest takes on special value, and this covered movable and immovable property forming part of the heritage which has greatest need for such protection" (Ley del Patrimonio Histórico Español, Ley 16/1985). The Assets of Cultural Interest (in Spanish Bien de Interés Cultural or BIC) encompass all tangible and intangible, movable and immovable manifestations that are witnesses to the historical and cultural values of a given society, in this case, Spanish society. The interest in surroundings or what is found around the monument first, then the asset of cultural interest, is also developed gradually, especially throughout the 20th century.

The first clear formulation on the protection of surroundings can be found in texts by Camilo Sitte (1843–1903) [1]. These ideas, considered by Gustavo Giovannoni (1873–1947) [2], took shape in the Athens Charter (1931), which recommended "that, in the construction of buildings, the character and external aspect of the cities in which they are to be erected should be respected, especially in the neighbourhood of ancient monuments, where the surroundings should be given special consideration." Therefore, monuments need their surroundings and character. The dismantling of the surrounding layout would constitute a loss for the monument itself.

The Venice Charter (1964), which is still widely referenced and present in the current culture of conservation, stated that the notion of monument "embraces not only the single architectural work but also the urban or rural setting in which is found the evidence of a particular civilization, a significant development or a historic event. This applies not only to great works of art but also to more modest works of the past which have acquired cultural significance with the passing of time".

The Quito Charter (1967), geared specifically towards the urban management of historic areas, focused its attention on rural and natural urban settings as assets of importance to heritage. The charter states that "since the idea of space is inseparable from the concept of monument, the stewardship of the State can and should be extended to the surrounding urban context or natural environment" including the cultural assets it contains while also pointing out that "the need to reconcile the demands of urban growth with the protection of environmental values is today an inflexible standard in the formulation of regulatory plans at both the local and the national levels. In this respect, every regulatory plan must be carried out in such a way as to permit integration into the urban fabric of historic districts and ensembles of environmental interest".

These topics were once again vehemently addressed in the context of legislation in the European Charter of the Architectural Heritage (1975), where the concept of architectural heritage (expanding the interpretation of monument) had already been established, through the statement that "the European architectural heritage consists not only of our most important monuments: it also includes the groups of lesser buildings in our old towns and

characteristic villages in their natural or manmade settings. For many years, only major monuments were protected and restored and then without reference to their environment. More recently it was realized that, if the surroundings are impaired, even those monuments can lose much of their character." Subsequently, these topics continued to be addressed in charters such as that of the Granada Convention (1985) and the Washington Charter (1987).

A major step was taken in the field of heritage analysis following the Convention for the Safeguarding of the Intangible Cultural Heritage (2003) where material cultural heritage (architectural, urban, movable, immovable, etc.), is awarded equal importance to intangible cultural heritage, which was defined as "the practices, representations, expressions, knowledge and skills—as well as the instruments, objects, artefacts and cultural spaces associated therewith—that communities, groups and, in some cases, individuals recognize as part of their cultural heritage". This new approach enriches the very concept of surroundings, as defined in the Xi'an Declaration (2005): "beyond the physical and visual aspects, the setting includes interaction with the natural environment; past or present social or spiritual practices, customs, traditional knowledge, use or activities and other forms of intangible cultural heritage aspects that created and form the space as well as the current and dynamic cultural, social and economic context".

All these reflections touch upon the essence of "surroundings", where it should be understood that there is no separation between the asset (or monument) and its surroundings (tangible and intangible environment), as the symbiosis between both does not allow their separation. The "monument" exists in relation to the surrounding tangible setting (urban, architectural, material, etc.), and the intangible one (historic, cultural, social, economic, etc.), which brought it about, and the surroundings or environment exist as the result of a culture (historic, urban, architectural, social, etc.), which has resulted in a series of tangible and intangible manifestations, some more notable than others, but equally valuable to the local culture.

### 1.2. State of the Art, Objectives and Methodology of the Research

From the late 20th century, and particularly the start of the 21st century, increasing numbers of studies have appeared on the architecture of the historic city of Valencia. These have also included several studies on the building of the Lonja de la Seda, detailing research on its history, architecture, construction and state of conservation (including [3–5]). Furthermore, research has also been published on the Plaza del Mercado, the Mercado Central and different markets in Valencia [6–11], as well as surrounding buildings (including [12,13]).

Publications were also found on urban development and urban planning in the city of Valencia during different periods (including [14–17], their archaeology [18,19], compendiums of views and historic engravings [20,21], and historic documents [22], as well as studies on important buildings from different periods in history [23–25]. However, previously the fabric of historic residential buildings had only been examined in a few select cases [23,26].

Given the limited number of detailed studies on historic residential buildings in the city, in the year 2000, the authors of this text began their research on dwellings in the historic city of Valencia, examining them from a historical, architectural, material, constructive, and conservation perspective. This study, developed over almost 15 years and published almost in its entirety in the two volumes of the book "Centro Histórico de Valencia. Ocho siglos de arquitectura residencial" [27] constituted an advance in its field and was recognised with different research awards. The façades of historic residential buildings within the perimeter of Ciutat Vella, or the historic centre of Valencia, considered an asset of cultural interest, were all studied and catalogued in detail, while the urban and architectural development of the city was also expanded on. Studies were also carried out on fully accessible buildings, as well as constructive materials and techniques (rammed earth walls, brick, walls, timber floors and ceilings, renderings and polychromy, joinery, balconies, building interiors, etc.). In addition, a filtering process was carried out on all the construction permit files from the 18th century and on a selection of 19th century files found in the Municipal Historical Archive of Valencia (AHMV onwards).

This allowed the authors to study both the layout of the buildings over time and any transformations observed over the last three centuries. Furthermore, laboratory studies were used to determine the composition of the materials, while dendrochronology was used to date timber elements, and chronotypology to date walls, floors, ceilings, joinery, ironwork, balconies and doors. This detailed material study and the dating of historic residential buildings were cross-referenced with the protection plans in place at the time in order to identify any shortcomings in the protection of these buildings [28]. Part of this study has been incorporated into the new protection plan for the city, which was passed in 2020 [29] (Figure 2).

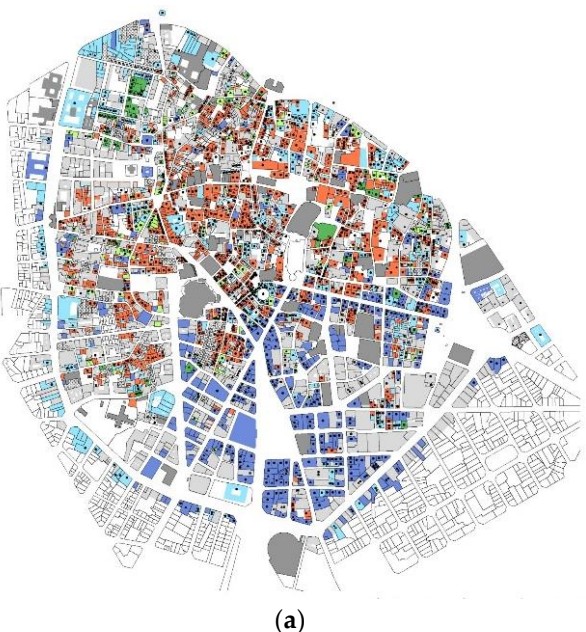
(**a**)

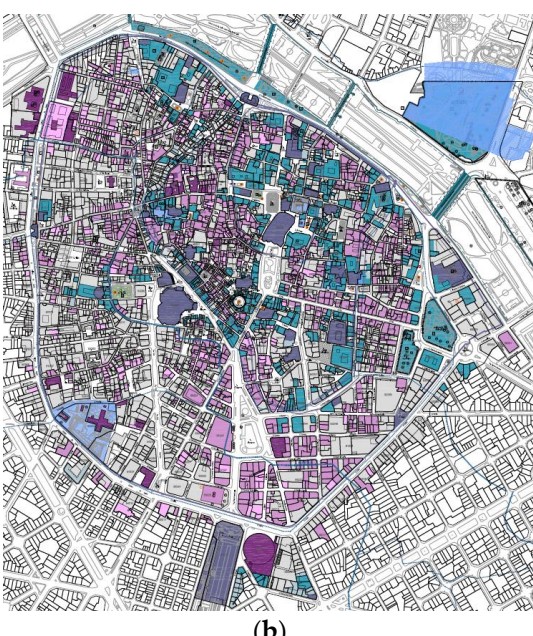
(**b**)

**Figure 2.** (**a**) Map by the authors, superimposing buildings protected until 2019 (with different shaped black symbols depending on the level of protection) and buildings deserving protection according to the authors' research (with different colours used to denote building characteristics). (**b**) Protection plan passed in 2020 including many of the buildings highlighted in the map drawn up by the authors [29].

As mentioned above, these studies generally focused on the area known as "Ciutat Vella", but no specific neighbourhoods in the historic city centre were studied in detail. This article presents later research that partly used already available data, adding new information to focus on a specific area outlined within the historic city of Valencia as a case study: the surroundings of the Lonja de la Seda. This new research aims to combine different historical, literary, artistic, documentary, material, constructive and urban perspectives in order to highlight the material, social, historical, urban, architectural and ethnological values of the historic centre of Valencia.

The case study chosen is the neighbourhood of the market, or in broader terms, the surroundings of the Lonja de la Seda in Valencia (Spain), declared World Heritage by UNESCO in 1996 (Figure 3). The subject of study in this research is not the Lonja as a monumental building of recognised heritage value, but rather everything that surrounds it. For this, a study was carried out on an urban scale on the evolution of the area, based on historic views and urban plans. The evolution of urban spaces such as squares and streets was analysed, while historical, literary and ethnographic analysis was also carried out on the published bibliography, views and historic photographs.

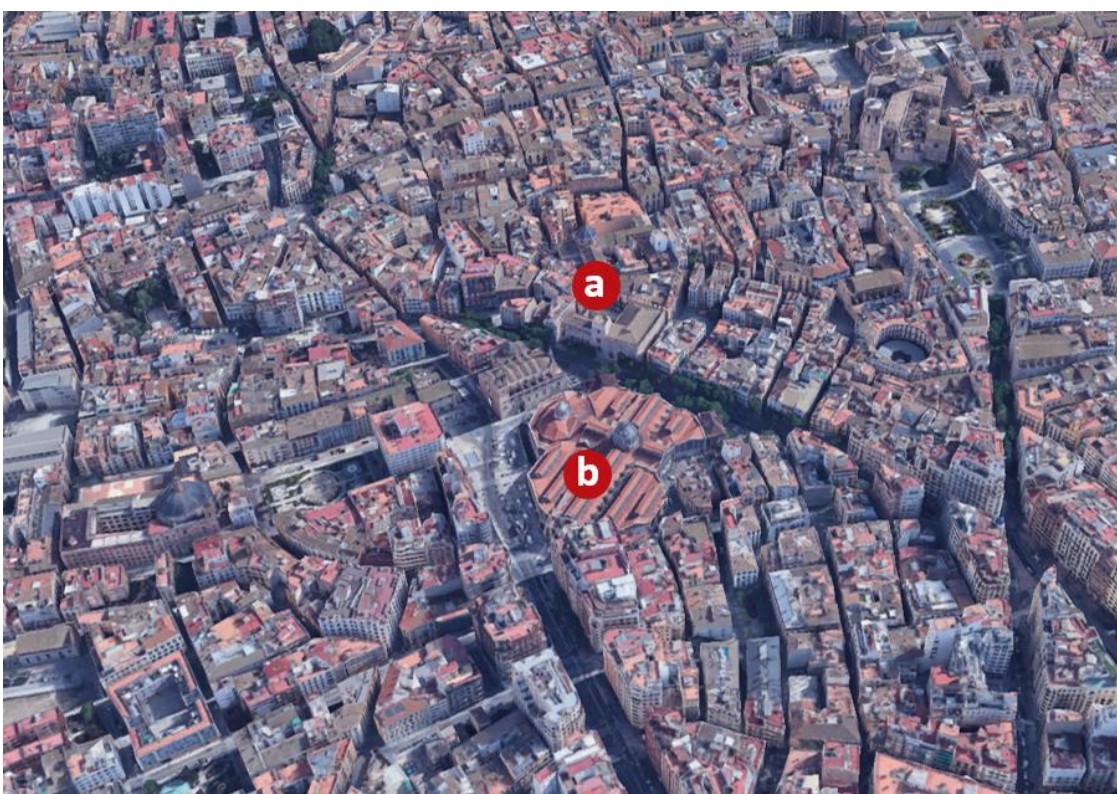

**Figure 3.** Aerial view of the surroundings of the Lonja de la Seda (**a**) and the Mercado Central (**b**) (Source: Google Earth).

Finally, a material and architectural analysis was carried out on all the monumental and non-residential buildings of the chosen setting, as well as on the residential buildings within the same area. For the purposes of this, analysis is divided into different elements such as squares, streets, emblematic buildings (markets, churches, palaces), dwellings or residential buildings. All these elements, which are analysed from historical, urban, architectural and ethnographic perspectives, provide a wealth of new data as well as a global approach offering a broad-ranging and multifaceted interpretation of the urban surroundings. This research was carried out through a review of the bibliography, the documentary study of archive sources (especially in the AHMV) and the direct study of buildings (architectural types, materials and constructive techniques used, transformations over time, etc.). The combination of several direct and indirect methods in this research has made it possible to identify the tangible and intangible values of a specific case that had not been studied previously—the surroundings of the Lonja—whose historical importance as a market place makes it an emblematic element of the city.

## 2. The Surroundings of the Lonja: Points for Consideration

In legal terms, the Special Plan for the Ciutat Vella district (passed in February 2020) outlines a perimeter for the surroundings of the Lonja, considered an asset of cultural interest (see fiche C1-13 from the Protection Catalogue; [29]). This perimeter (Figure 4) includes notable buildings such as the Lonja and the adjacent Consulate of the Sea (Figure 4b(a)), Mercado Central or Central Market (Figure 4b(b)), and churches such as those of Santos Juanes (Figure 4b(c)) and Sagrado Corazón de Jesús de la Compañía or Jesuitical church (Figure 4b(d)); public spaces such as the squares of Mercado (Figure 4b(e)), Collado (Figure 4b(f)), Ciudad de Brujas (Figure 4b(g)), of la Compañía (Figure 4b(h)) and some streets and avenues such as María Cristina avenue, and the streets of Taula de Canvis, Cajeros, Danzas, de la Lonja, Pere Compte, Ercilla, Derechos, Vieja de la Paja, Sampedor; as well as the dwellings and buildings included in or adjoining this perimeter.

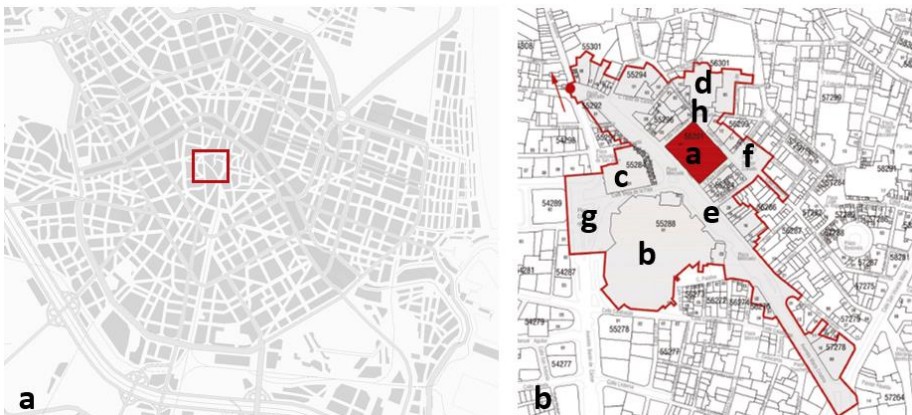

**Figure 4.** (**a**) The location of the Lonja de la Seda and its surroundings in the whole of the city of Valencia (red square); (**b**)delimitation of the protection environment of the silk market (identified in red) (source: (**a**) authors; (**b**) [29]).

However, beyond the legal definition of surroundings for assets of cultural interest, and with the freedom afforded by not having to establish regulations, this text aims to analyse certain elements or factors which over time have contributed—and continue to contribute—to the definition of the surroundings of the Lonja. Most of the images of the Lonja produced by 19th century travellers (including Laborde, 1807; Charles de Lalaisse, 1812; Rouargue, 1850) [20,21] show the market square on the right-hand side of a triangular space, where it appears surrounded by residential buildings, opposite the church of Santos Juanes and on occasions the market building. The square is always bustling, full of people and stalls. The view of the 19th century traveller is fixed on the elements that define the surroundings while also providing guidance (Figure 5).

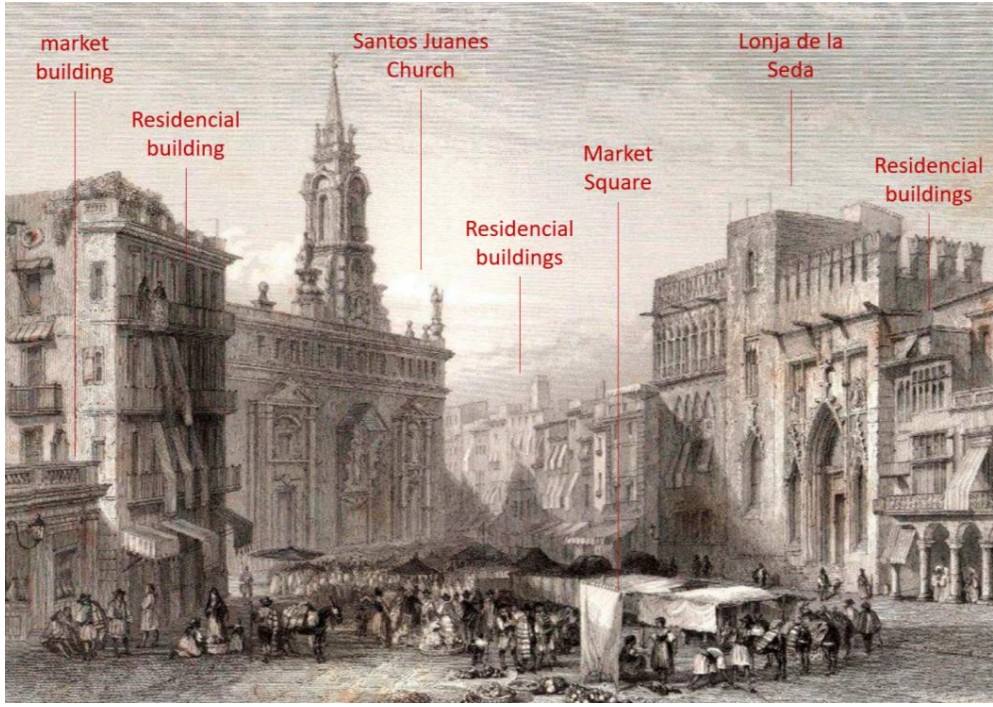

**Figure 5.** View of Plaza del Mercado (author: Rouargue, 1850 [21]), showing the Lonja de la Seda to the right, the market buildings and the Church of Santos Juanes to the left, and the surrounding residential buildings characteristic of the square's surroundings.

*2.1. The Market Square*

The Plaza del Mercado or market square in Valencia is an outstanding example of the urban geographical inertia of a function, in this case, the buying and selling of foodstuffs and objects. This is not unlike the inertia of the position of a religious building within a city. As the centuries pass, the container for a function can be replaced or transformed but the activity remains. This is what has happened with the site of Valencia Cathedral, which was previously a mosque, before that a Visigoth basilica, and initially, a Roman temple. The market square has been fulfilling the same function for at least a thousand years.

The market square is without doubt one of the most important elements of the surroundings of the Lonja: a space among buildings that is empty, yet always full of activity and people. Over centuries this square has been shaped and gradually transformed thanks to the new forms of design observed in cities and public spaces, as well as different historical developments and new demands. The market square was defined fundamentally during the Middle Ages, based on the profile of the Moorish town wall on its northeast side. The area occupied by the square has never had a clear and precise geometry. Its outline is the result of the construction of buildings that throughout history have progressively drawn these limits: the churches and convents, the markets, the Lonja and the buildings with shops and dwellings. Therefore, the form and dimension of this square have depended more on its outer limits than on the space itself and, as a result, both its shape and surface have changed over time.

The space currently occupied by the market square was originally outside the walls of the Islamic city of Balansiya (*arrabal* of Boatella), between the Gate of Boatella (*Bab al-Baytala*) and the Gate of the Alcaicería (*Bab al-Qaysariya*). The Gate of Boatella must have been one of the main accesses to Balansiya, used for the passage of goods. Furthermore, near the Gate of the Alcaicería there was market activity, both inside and outside the city walls ([18], p. 425). The market area in the city of Balansiya could therefore be said to have been consolidated after the city's conquest by Jaime I in 1238 and was in the area once occupied by the market of the Islamic city. Outside the wall, opposite the Gate of Boatella, was the *arrabal* of Boatella, a small rural settlement near a large necropolis dating to late antiquity [19]. This *arrabal*, made up of houses inhabited at the time of the conquest of Jaume I and referenced in the *Llibre del Repartiment,* may have been walled, with two accesses, as gleaned from the information from Barceló [15].

Although in the Islamic era, there was already a space outside the walls used as a market, the consolidation and development process began in the Middle Ages, when the space outside the walls was taken over through the construction of convents, the incorporation of the *arrabales* and the foundation of the *pueblas* (medieval urban settlements), which gradually formed the outer perimeter of what later became a medieval city delimited by walls erected throughout the 14th century [30]. This process of progressive colonisation of the territory surrounding the medina led to the definition of the market square through the gradual construction of a number of religious buildings. The foundation in 1240 of the Convent of La Merced ([8], p. 72) was followed by that of the Convent of Santa María Magdalena, and the establishment in 1268 of the former hermitage of Santos Juanes. This space delimited by religious buildings to the southwest and by the Islamic wall to the northeast was gradually incorporated into the medieval city, where increasingly bustling temporary and permanent shops could be found in the triangular area, while artisans and workshops tended to be located on the perimeter. The construction of the new medieval wall and the sizeable expansion of the urban area meant that the market square remained within the geometrical centre of the city.

On the night of 16 March 1447, for a full seven hours, a fire raged through almost two hectares of houses on a corner of the market square. This event was recently studied in depth to document information relevant to the history of this part of the city [31]. The fire, which was probably started as revenge for an execution that had taken place in the same square days earlier, destroyed 46 houses and the fish stalls of the market and resulted in the death of 10 people. According to reports at the time, the fire completely destroyed the

carpentry workshop as well as the buildings from the cemetery of Santa Catalina to Puerta Nova, and from El Trenc to the poultry and fish sellers. As Jaume Roig ([16], p. 77) put it at the time: "...La Pellería, Trench-Fusteria, Fins mig mercat, Nas vist cremat."

The number of people receiving damages for the destruction caused by the fire provides a very vivid picture of the range of artisan activities occurring in this district—and so near the market square: carpenters (the guild most affected by the fire), merchants of wool fabric, spices or second-hand clothes, bakers, cobblers, ironmongers, tailors, painters, cheesemakers, box manufacturers, haberdasheries, etc. Furthermore, following the extensive fire damage to the buildings, a plan for urban regularisation was implemented to rectify the existing streets and restructure the area ([31], p. 514). No images of Valencia at this time are conserved, but it is apparent that these works to straighten the streets around the square aimed to improve and consolidate a part of the city that was gaining importance and becoming increasingly representative, a status finally cemented between 1482 and 1548 with the construction of the new Lonja de la Seda.

Valencian humanist Juan Luis Vives (1492–1540), who lived in the city until 1509 and knew the market square, was able to see the completed Contract Hall of the Lonja and the adjacent Consulate of the Sea under construction. Away from his family and city, through his literary alter ego Centelles, he said: "What a large market! What excellent order and distribution of sellers and goods! The scent of these fruits! The variety! What beauty and cleanliness! No orchards can compare to those that supply this city, nor is there diligence to equal that of the market inspection and his ministers so that buyers are not deceived ..." ([32], p. 356).

The first existing view of Valencia is that drawn by Anthoine Van Den Wijngaerde (1512/1525–1571) in 1563 [20]. This view shows—albeit unclearly—the market square and the surrounding representative buildings: the religious building on the left, with the lettering "Madalena", is the convent of Santa María Magdalena; in the centre, the caption "Logia" refers to the space at the back of the Lonja de la Seda, possibly as a way to distinguish between the Lonja de la Seda and the Lonja del Aceite (Oil Exchange), which may be the building bearing the same lettering and seen on the left; on the far left, with the caption "S. Joan" we find the church of Santos Juanes; in the centre of the square the author draws a structure that is identified as the gallows ([8], p. 93).

On the plan drawn up by Antonio Mancelli (?–1645) in 1608, in the perimeter of the market square we see the notable buildings of the Lonja de la Seda (Figure 6a(1), the church of Santos Juanes and the convents of La Merced and Santa María Magdalena. Behind the Lonja de la Seda, Mancelli draws attention to the Lonja del Aceite (Figure 6a(2). Mancelli's plan shows the market square as a vast empty space with a gallows in the centre which brings to mind the public executions which took place there. The simple profile of the church of Santos Juanes (Figure 6a(3) is highlighted as it shows the church prior to the remodelling work, which resulted in the new church façade looking onto the market square, as seen in the 1704 plan by Father Tosca. However, the church depicted by Mancelli does not blend in with the square and appears completely disconnected from it, despite helping to define the northwest limit. Nevertheless, the edge of the square appears to be delimited by the notable buildings, especially by residential ones, which are anonymous, and follow the same model of ground floor access and first-floor windows. The author provides a clear simplified representation showing the same pattern of residential buildings that is repeated on every plot: a ground floor door and two first-floor windows.

The following plan of the city of Valencia was drawn by Father Vicente Tosca (1651–1723) in 1704, almost a century after Mancelli's (1608). On Tosca's plan, and in the updated version by José Fortea (1738), it is possible to see the transformations undergone by the market square in the 17th century, a topic widely studied by García Peris ([8], pp. 113–154): the construction of the chapel of Communion of the church of Santos Juanes (1643) (Figure 6b(4); the reconstruction of several collapsed houses in different parts of the square (1663–1666); the addition of the fountain by Juan Bautista Pérez Castiel (1650–1707) in the centre of the square (1672), which remained there until it was replaced by a larger

one (Figure 6b(5); the consolidation of the church of La Merced and the remodelling of the cloister (both completed in 1662), as well as the completed construction of the bell tower (finished in 1670) (Figure 6b(6); the demolition of the old church of the convent of Santa María Magdalena (1636) and the construction of the new church (completed in 1679) (Figure 6b(7); the Baroque updates to the church of Santos Juanes and the construction of the façade depicting scenes towards the square (1693–1702), including the construction of the podium which hosted *les covetes* (commercial premises) (Figure 6b(3)—expanded in 1713 to surround the chapel of Communion—thus creating 19 premises for vendors (seen on the 1738 plan). Finally, the appearance of porticoes on the different façades of the blocks that look onto the square should also be noted (Figure 6b(8)) on the buildings adjoining the convent of La Merced, on the two blocks between the convent of Santa María Magdalena and the church of Santos Juanes, as well as on the block found beside the chapel of Communion of the church of Santos Juanes.

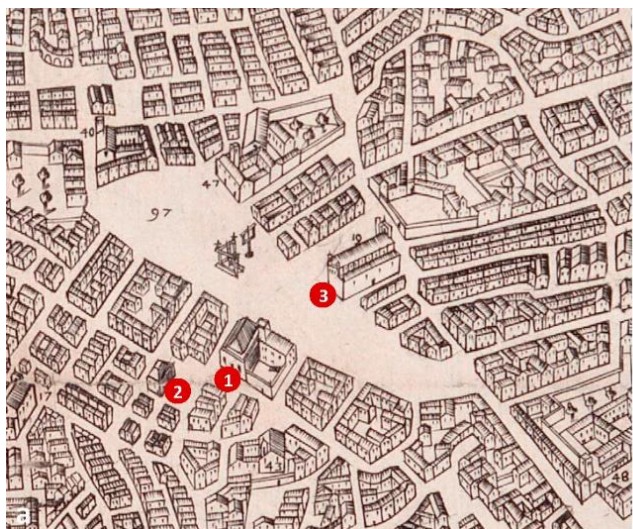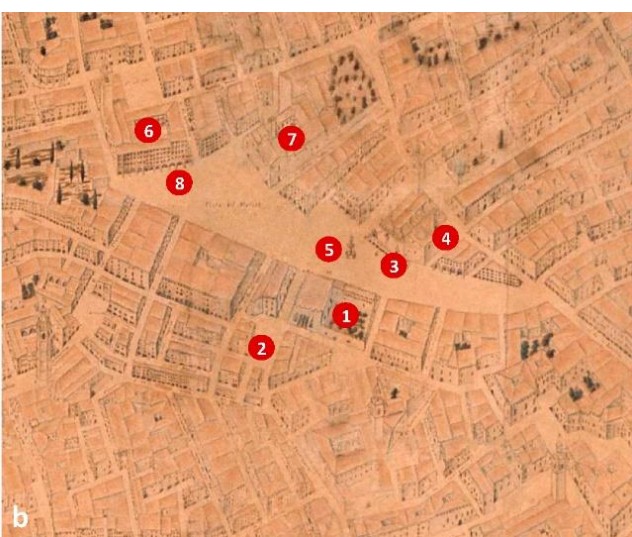

**Figure 6.** (**a**) Detail of the Market Square and the surroundings of the Lonja on the map of Mancelli (1608) [14]; (**b**) Plano de Tosca (1704) [14]; 1. Lonja de la Seda; 2. Lonja del Aceite; 3. Church of Santos Juanes; 4. Chapel of Communion of the church of Santos Juanes; 5. fountain by Juan Bautista Pérez Castiel; 6. church of La Merced; 7. convent of Santa María Magdalena; 8. porticoes on the different façades.

In addition, the ground floors of the buildings around the square are shown on 18th century plans with continuous doors which never fully become porches but serve as shop windows for businesses. The porticoes of the market square are shown on several occasions throughout the 19th century on plans and in views by different authors. In the "Plano geométrico de la plaza de Valencia y sus contornos" by Francisco Cortés y Chacón 1811 ([14], p. 44–45) the buildings with porticoes on either side in the market square are marked. These can also be seen in the view by Laborde around that time, as well as in later ones such as that by Aulaire in 1830 ([21], pp. 130–131).

After the confiscation of Mendizábal (1836), the market square was modified extensively following the demolitions of the convents on the southern limits of this urban space: the convent of La Merced and the convent of Santa María Magdalena. In 1839, the site formerly occupied by the latter saw the inauguration of a new market called Mercado Nuevo or Mercado de los Pórticos (new or porticoed market). The plot of land that had housed the demolished convent of La Merced was used for the construction of dwellings, which replaced both the convent and the housing with porticoes that had closed off the square on its south border.

These construction projects marked the start of a progressive modification of the urban space which could be described as "from the square to the street" (Figure 7), the

empty space by the market buildings was gradually occupied, first by the porticoed market (Figure 7b) and later by the central market, as well as housing. With the construction of the central market (Figure 7c), the market square was reduced to a series of expansions on a single street, taken up for some time by the traffic from carriages, the passing trams and vehicle traffic (Figure 7d). These almost residual triangular spaces completely blurred the concept and use of the former square, which had become unnecessary from the moment the market stalls were relocated to the central market. In this respect, it should be stressed that the project by the architects Peñín and Quintana [33] envisages the pedestrianisation of the market square and a more homogeneous and dignified treatment of this setting, which had gradually deteriorated, and clearly required a solution more in keeping with the heritage importance of the place. However, despite the elimination of pavements in the square, the market square remained a street with expansions, having lost its tangible and intangible dimension as a square over a century earlier.

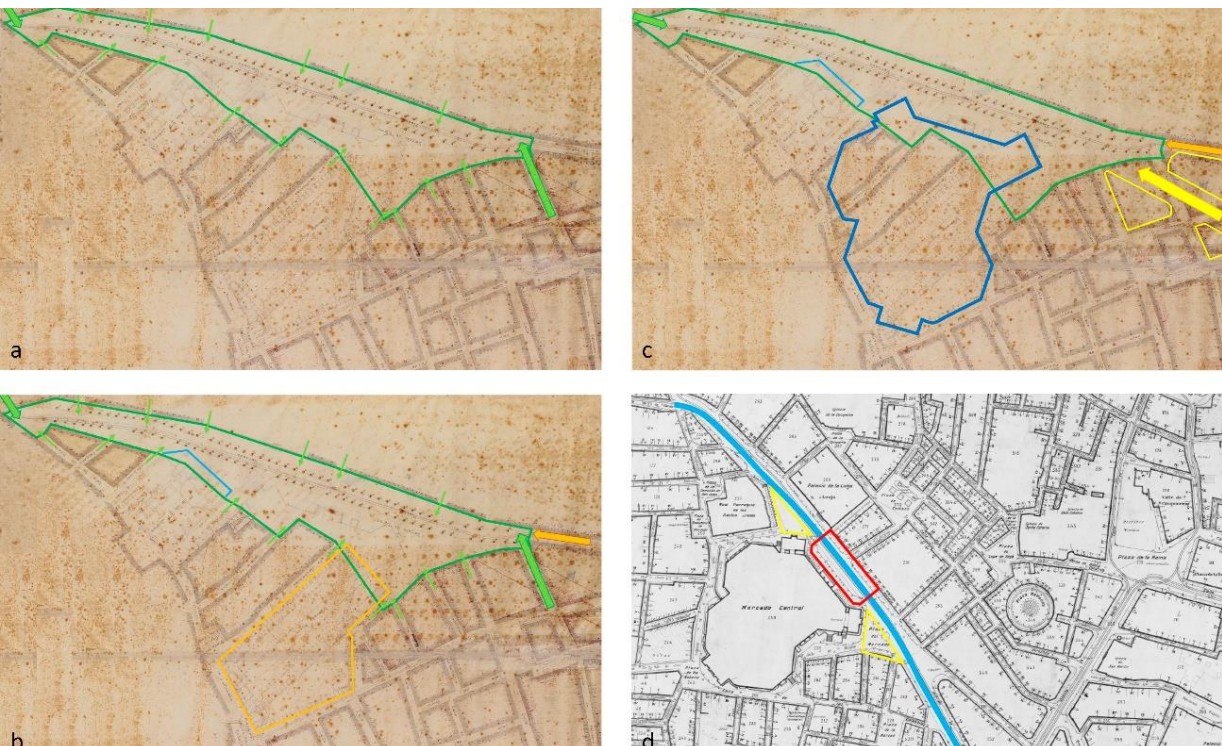

**Figure 7.** The evolution of the market square (the colours correspond to the different phases): (**a**) the square and its entrances before the transformations of the 19th century (in green); (**b**) the insertion of the Market of the Porticos (on the plan of 1892, [14]) (in orange); (**c**) the insertion of the Central Market (on the plan of 1892, [14]) (in blue); (**d**) the spaces (in yellow and red) resulting from the transformations of the square and the insertion of the tram (in cian) on the 1944 plan [14].

### 2.2. Spectacles of Life and Death

The market square, as the main public space in a city where it was once the only square, was the setting for a large number of events, demonstrations and feasts. For example, in 1606, on the first centenary of the canonisation of San Vicente Ferrer, Vicente Colomer designed a mannequin of the saint and made it fly from one end of the market to the other to the amazement of the crowds at what was probably the first zipline in Valencia ([6], p. 144). While the celebration of bull runs, student riots, tournaments and jousting battles was also common, the best spectacle in itself was the daily hustle-and-bustle of the market. In 1494, the traveller Hieronymus Münzer (1495) wrote [34]:

"The inhabitants of the city, both men and women, walk around through the streets at night, and there are always so many people that it looks like a fair is being held (. . .) I

would never have believed that such a spectacle could exist if I had not seen it myself, in the company of my fellow countrymen, the honourable merchants of Ravensburg. The food shops do not close until midnight, so one can buy whatever one wants at any time."

In addition, the function of the gallows, clearly visible in the market square, was to act as a deterrent to prevent inhabitants from carrying out any crimes. Traditional punishments, even for minor crimes, were terrible: lashings; amputation of body parts such as hands, ears, or arms; exile; and even death. Offending the clergy, for instance, was punishable by nailing a hand to wood.

The executioner or *morro de vaques* was a civil servant who was paid a given rate for each of these actions. Although these punishments were applied to all social classes, it was easier for wealthier individuals to evade corporal punishment or the death penalty. According to reports, in 1524 a public gallows with three stone pillars was built so that the provisional timber structure in use there since at least the 14th century, and set up for executions, would not collapse when the hanged prisoner was dropped. In 1599, it was dismantled for the wedding of Felipe III in Valencia Cathedral and was rebuilt shortly afterwards, appearing in Mancelli's plan. In 1632, it was dismantled once again due to King Felipe IV's passing through Valencia ([35], p. 14–19).

The bodies of the dead were sometimes left hanging for hours, or till the next day, to serve as a public example. Although this was to set an example, market stall vendors disapproved as they felt this affected their business negatively. In the 15th century, Jaume Roig ([36], pp. 530–533) stated: "*nor I would eat meat at the market if there were any man was hanging there*".

The dead prisoners were buried in the sector for hangings in the cemetery of Santos Juanes and later by the ravine of Carraixet ([6], p. 67). In some of the more flagrant cases coming from the Inquisition Tribunal, not only was burial denied, but the bodies were left to be eaten by dogs. This was the case with the last victim of the Inquisition, Gaetà Ripoll, a deist schoolmaster from Ruzafa who was sentenced on 31 July 1826, following a lengthy trial in which he was accused of not going to mass on Sundays, only teaching students the commandments of the Law of God, making them use "Loado sea Dios" ("Praise be to God") as a greeting instead of "Ave María", and not having taken them to adore the viaticum as it passed the school. As a concession to the spirit of the times, he was not burned on the pyre but hanged in the market, over a barrel symbolically painted with flames. After the hanging, he was moved from the gallows at the market to La Pechina from where he was thrown onto the dry riverbed to be eaten by vermin ([17], p. 429).

Prosper Mérimée (1803–1870) made seven trips around Spain between 1830 and 1864 ([37], p. 36). On the first of these, he witnessed an execution in the market square:

"The square was far from full. The fruit and vegetable sellers had not moved from their stalls. It was easy to move around everywhere. The gallows, topped with the Aragon coat of arms, stood opposite the Lonja de la Seda, an elegant building in the Moorish style. The market square is long. The houses which form it are narrow, with several storeys, and each line of openings has a balcony with iron bars. Seen from afar they resemble huge cages. On many of the balconies with bars, there were no spectators. This indifference may be the result, perhaps, of the industrious idiosyncrasy of the Valencian people" ([38], p. 57).

### 2.3. The Fountain

Whereas a decorative fountain had spouted wine in the market square in 1585 for the visit of Felipe II ([6], p. 159), in 1672 Juan Bautista Pérez Castiel built the first true fountain in the market square, with water driven by a wheel located on Cenia Street (Figure 8a). It was also used to water the garden of the Lonja and supply water to the Guild of Water Carriers or firemen, whose headquarters were in the basement of the adjacent Consulate of the Sea ([6], p. 68). This fountain was replaced in 1852 with a cast iron fountain manufactured in France (Figure 8b). This fountain, which arrived in Valencia by boat, was transported for free from the port to the city centre on the recently inaugurated railway, whose owner, José Campo (1814–1889), had been born at number 80 of the market square ([6], 190) and

wished to improve his neighbourhood. In 1875, some sculptures of children and decorative details were added. In 1878, this fountain was moved to Alameda Avenue, beside Mar Bridge, before being moved again in 1933 to its current location, also on Alameda Avenue, beside Aragón Bridge (Figure 8c).

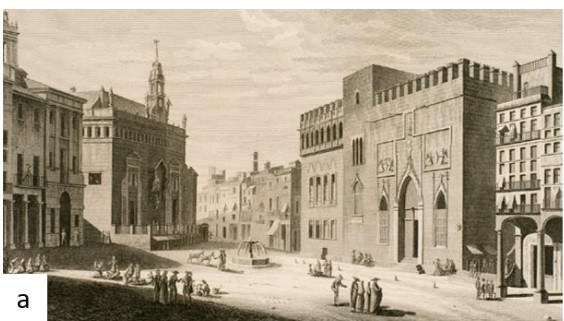
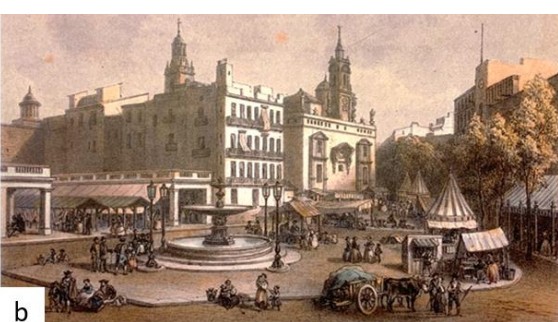
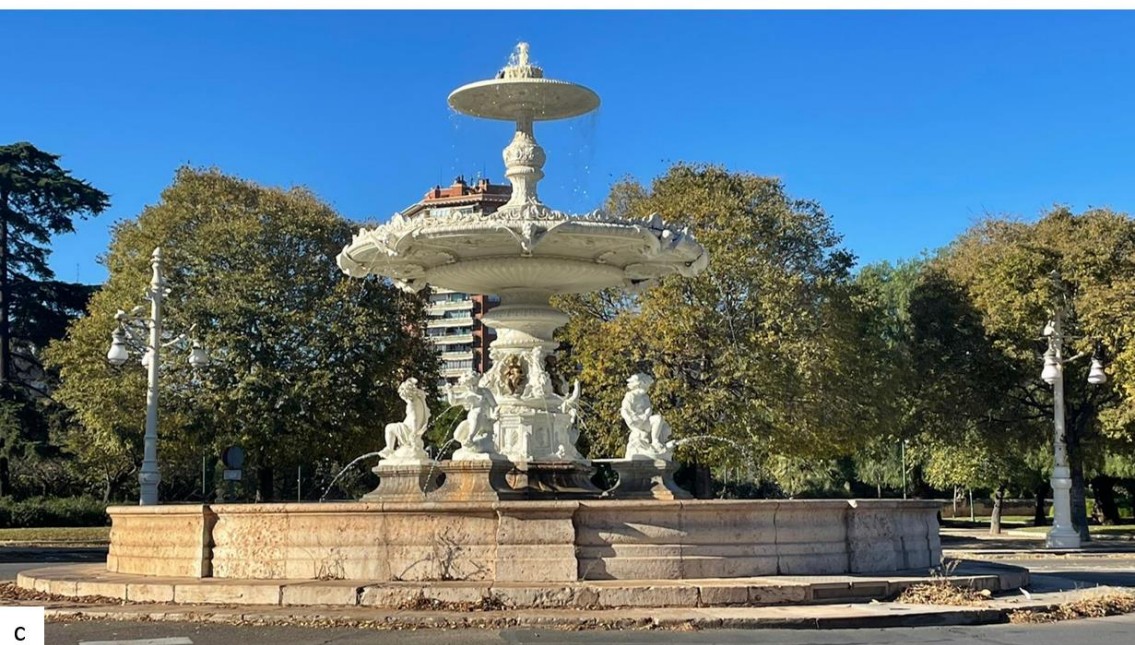

**Figure 8.** (**a**) The first true fountain in the market square built by Juan Bautista Pérez Castiel in 1672 (drawing A. Laborde, 1811 [21]); (**b**) the cast iron fountain manufactured in France that replaced the first fountain in 1852 [21]; (**c**) the fountain in its current location on Alameda Avenue (source: authors).

### 2.4. Old Lonja or Lonja del Aceite

This building, which has now disappeared, stood on the current Collado Square and preceded the Lonja de la Seda. It fitted the typology of loggias of the Kingdom of Aragon or Northern Italy, with porticoes on the ground floor and a contract hall on the upper one. The portico was probably built using stone ashlar, while the upper floor was almost certainly brick-supplemented rammed earth. Probably built before 1314, it was successively expanded in 1346 and 1444, and shortly afterwards two atlantes were placed on the corners and became known popularly as Engonari and Engonariesa. Its ground floor archway on three sides had bars added in 1377 and was partially closed off with doors in 1734 to prevent the mess that tended to accumulate inside. On the roof, opposite Derechos Street, stood a stone throne used to display criminals for their public shaming [39].

Once the Lonja de la Seda was built, the Lonja del Aceite was known as Old Lonja, as opposed to the New Lonja. It was later called the Lonja del Aceite or Llotja de l'Oli, although it did not only sell oil but also honey, flour and almonds. The building was so old that it was even referenced in popular sayings: "It is even older that the Lonja del

Aceite" ([6], pp. 39–40). It also hosted the market inspection, which until 1372 was located at Trench Street. Later, it was based in the Longeta del Mustasaf in Santa Catalina Square until 1594, before being moved again, to the Lonja del Aceite, where it remained until 1838. It was then situated in the market square, beside the former porticoed market, until 1916. The Lonja del Aceite was demolished unexpectedly in 1877 and today, only an olive tree planted in Collado Square evokes its presence and disappearance.

### 2.5. Church of Santos Juanes

This church was founded, following the reconquest of Jaime I, on the site of a mosque outside the city's Moorish walls, yet another example of urban geographical inertia, like the market. This primitive medieval construction was rebuilt and remodelled on several occasions following the fires of 1311 and 1592, the construction of the Chapel of Communion in the mid-17th century and the transformation of the interior vault and construction of a new façade looking onto the market square in the late 17th century [25,30]. Furthermore, in 1700 the church of Santos Juanes applied to the Council of the City of Valencia for the cession of land behind the church, planning to build a terrace to cover the market stalls that had occupied this space since the time of Jaime I. Once the permit was granted on 1st August 1700 ([12], pp. 11–12) the sculptor Leonardo Julio Capuz Calvet (1660–1731) was put in charge of its construction in exchange for 67 annual rents as payment for the construction of the market stalls, their doors and the upper terrace. Eighteen years later the sculptor sold this usufruct to the parish of Santos Juanes ([40], p. 60). During the work, numerous burial sites belonging to the Moorish cemetery of the former mosque or *almacabra* were unearthed. Father Tosca included this terrace with small premises on its lower perimeter in the 1704 plan. These were once known as *les paradetes de Sant Joan* or *les llanterneries* but are currently known as *les covetes de Sant Joan* (Figure 9). The construction of this terrace took up approximately 110 m$^2$ of public land from the market square.

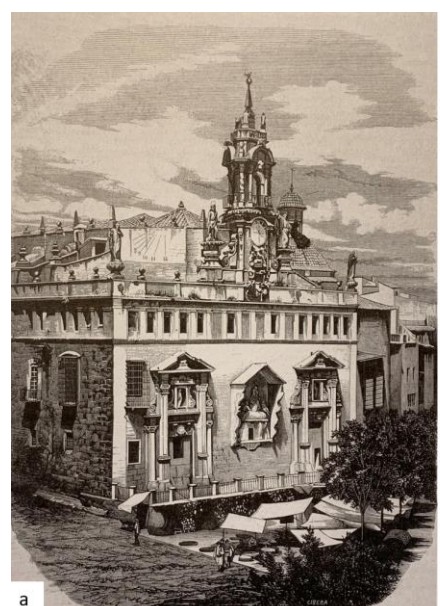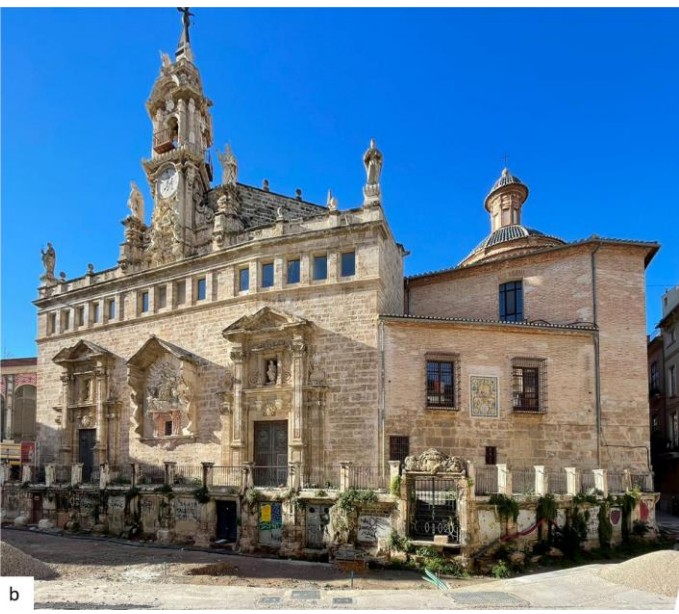

**Figure 9.** *Les covetes de Sant Joan*: (**a**) *les covetes* in an 1857 engraving [21]; (**b**) the excavation carried out to bring to light *les covetes* (2021) (source: authors).

It is striking how the increase in the level of the market square over the last 325 years has led to these market stalls becoming semisubterranean. Apparently, this was not the case initially. Recent excavation carried out in the market square for the new paving shows that the bases of the stone ashlar pilasters between the stalls were definitely designed to be seen. Based on the engravings and photographs of the steps to the right of the Lonja, and the descriptions by Llombart [41], it is thought that the level of the square has probably

increased by 70 cm since 1700—and probably more going back to 1484—although the increase in height of the paving would have been slower initially and accelerated from the mid-19th century onwards.

This façade of los Santos Juanes and its magnificent terrace, from where the spectacle of life, death, monuments and the space of the market square could be contemplated, suffered extensive damage in the bombing orchestrated by Capitán General Rafael Primo de Rivera y Sobremonte (1813–1902) in the second week of 1869 against the most densely populated areas in the city, as military repression against the federal revolution. Renowned American journalist Henry Morton Stanley (1841–1904), a war correspondent for the New York Herald in Valencia, described it:

"Miraculously, the Gothic Lonja has been spared from damage. However, this has not been the case with the church of Santos Juanes, located opposite. A statue has been knocked over; the niche of the Virgin has been vandalized and defaced; the gargoyles have lost their stone wings, and the church tower has also suffered extensive damage (...) A dozen trees have been cut down in the market. Eight houses there have been rendered unusable; they will have to be rebuilt. The market itself has suffered greatly: collapsed columns, a broken roof, destroyed stalls, etc." ([42], p. 158) (translation by the authors).

In the space of only fifteen minutes, Stanley counted up to one hundred and thirty barricades around the market square. The position of these barricades, put up by Milicia Nacional volunteers and the army of Primo de Rivera, and their evolution in this mini-civil war, which took place between the 8 and 16 October was documented, as can be seen in two plans of Valencia from 1869 ([14], pp. 84–85, 90–91). Stanley also mentioned the existence of graffiti on the Lonja, which at the time was crowned by a flag saying, "Federal Republic", and the wall which closed off the Escalones de la Lonja street: "War on the general and peace to the soldier" and "Death penalty to the thief". The description of the market square following this major battle, combined with the trademark literary style which Stanley imprinted on his chronicles, is chilling:

"What a foul stench! What disgusting stains! What a smell of blood! The blood appears in small puddles, in putrid streams, giving off deadly miasmas. As for the marks of war, it is enough to see the bullet-riddled market stalls. Observe the destroyed balconies, the chipped balustrades, the broken brackets. Look at the walls full of scars. How strange the medieval figures, with their cut-off heads and golden crowns! Even the Virgin has been sacrilegiously and irreparably mutilated, and the Santos Juanes look as if they have been shot for high treason (...) But the scars of the harsh location are far too numerous to list. They are everywhere; they become visible in the ruin and damages of the square, in the fallen trees which once provided shade in the market square, in the horrendous ruins of the surroundings" ([42], pp. 157–158) (translation by the authors).

Twelve days later, on 28 October, Stanley was already in Paris when he received the task of locating and interviewing the missionary David Livingstone (1813–1873). Towards the end of 1871, Stanley found him in Ujiji (Tanzania), where he remained with him until March 1872. A year later, Stanley, a global celebrity at this point, returned to Valencia as a war correspondent to report on the disturbances brought about by the cantonal revolution of Valencia in August 1873, although in his opinion these revolutionaries paled in comparison to those of the federal revolution of 1869.

From its magnificent 300-year-old vantage point, the church of Santos Juanes witnessed and, as seen, was even a victim of, historical developments. Unfortunately, the fact that this vantage point and possible access to the church are now unusable—partly due to the transformation from an urban historic square to a street with expansions—is sadly a wasted opportunity, preventing it from becoming part of the setting of the market square, as well as of the comings and goings of residents and tourists.

*2.6. Panses Square, Today Called Compañía Square*

This building, located behind the Lonja, was also witness to major episodes in the city's history. Built between 1595 and 1631 on a site specifically selected by San Francisco de

Borja (1510–1572) for his foundation, it appeared to still be under construction on Antonio Mancelli's (¿?–1645) plan from 1608. The Jesuitical professed house or adjoining building in Cenia Street was built between 1668 and 1669 and, as a result, Father Tosca's (1651–1723) plan faithfully reflects the full layout. In 1767, following the expulsion of the Jesuits from Spain on the orders of Carlos III (1716–1788), it seems the church was left half-abandoned and the Casa Profesa was used as the Archive for the Kingdom between 1810 and 1963 [13].

The name of this square, known traditionally as Panses Square, as it was where most of the raisins were sold, eventually changed to Compañía Square. It was customary to read the daily press there. A plaque on the back façade of the Lonja commemorates the 23 May 1808, when the people of Valencia declared war on Napoleon, supporting the command of the *palleter* Vicente Doménech, spurred by indignation at the news published in *La Gaceta* on the abdication of King Fernando VII and the advances of French troops in Spain. The profession of *palleter* consisted in selling sulphur wicks, lighting them to disinfect the empty wine barrels and prevent the proliferation of microorganisms which might alter the wine.

Sixty years later, when the Revolution of September 1868—known as "la Gloriosa"—led to Queen Isabel II (1830–1904) being deposed and exiled, the Junta Superior Revolucionaria de Valencia, headed by the former mayor José Peris y Valero (1821–1877), ordered the demolition of the Jesuitical church [43]. Years later, in 1885, the architect Joaquín María Belda Ibáñez (1839–1912) built a new church, still seen today, following the outlines of the demolished church. The residential buildings on the corners of Lonja Street and Cenia Street and Cordellats Street and Danzas Street, examples to be used for in-depth analysis, have witnessed—as is the case with other buildings—part or all of what has happened in the history of the square or the Lonja itself.

The building at Compañía Square n. 3 on the corner of Cordellats Street is of great interest (Figure 10). This three-storey building is made up of a ground, first and second floor. At first glance, the dressings of the windows on the first and second floors and the railings of the balconies appear to date from the early 20th century. However, this is merely the appearance. In fact, the railing of the first-floor balcony, with its Modernist decoration and lower frieze with bees representing hard work, is made of cast mould, and the solid base of the balcony probably dates from the same period, despite the possibility of a more recent intervention. The balcony windows on the first floor, looking onto Danzas Street, still conserve the valances of die-cut wood board used to protect the external solar protection blinds of the building, another characteristic feature of the Valencia of the time. The wooden joinery, with an upper clerestory and interior shutters, may also date from the same period. In fact, the iridescence of some of the glazing of the clerestories reveals that they were early 20th century blown glass.

However, the upper balcony with rounded corners is much older (Figure 10b), as can be seen from the ceramic tile on the underside of the balcony on Danzas Street, made up of pieces measuring half a palm (11.25 cm), a measure for ceramic production which disappeared around 1740 [44]. A more detailed examination reveals that the railings are wrought iron from a square section and that the lower protective band with flowers framed in rhomboids and a central ring with scrolls, also wrought iron, were subsequently added with small rivets, probably at the time when the lower Modernist balconies were added. It could be assumed that the dressings of the second-floor windows are not Modernist, but date from the 18th century Baroque, although at that point the artisan dwellings were not decorated, far less so on the top floor.

But there are other striking details (Figure 10c): the ground floor in rough ashlar, with a simple corner without decoration and a Roman arch at the entrance. This suggests it was a medieval construction built before the War of the Guilds (1519), after which Roman arches were rarely found at the entrance of residential buildings. The brick construction visible in sections missing rendering is in brick-supplemented rammed earth in dimensions characteristic of the 14th century. In contrast, the industrial brick used to block off the Roman arch is characteristic of the late 19th or early 20th centuries, which is in line with the dates of the interventions on the first-floor balconies. The somewhat irregular distribution

of the openings on the ground floor, the windows on the first floor, and the loggia on the second floor with no proposed vertical symmetry, also suggests a building prior to the advent of the Academia de Bellas Artes de San Carlos, which sought geometric order on façades whenever possible. In addition, as will be detailed later on, the 18th century saw the start of a process of completing medieval crown loggias, which were transformed as yet another floor on the building in order to accommodate the city's growing population, a process not observed on this building.

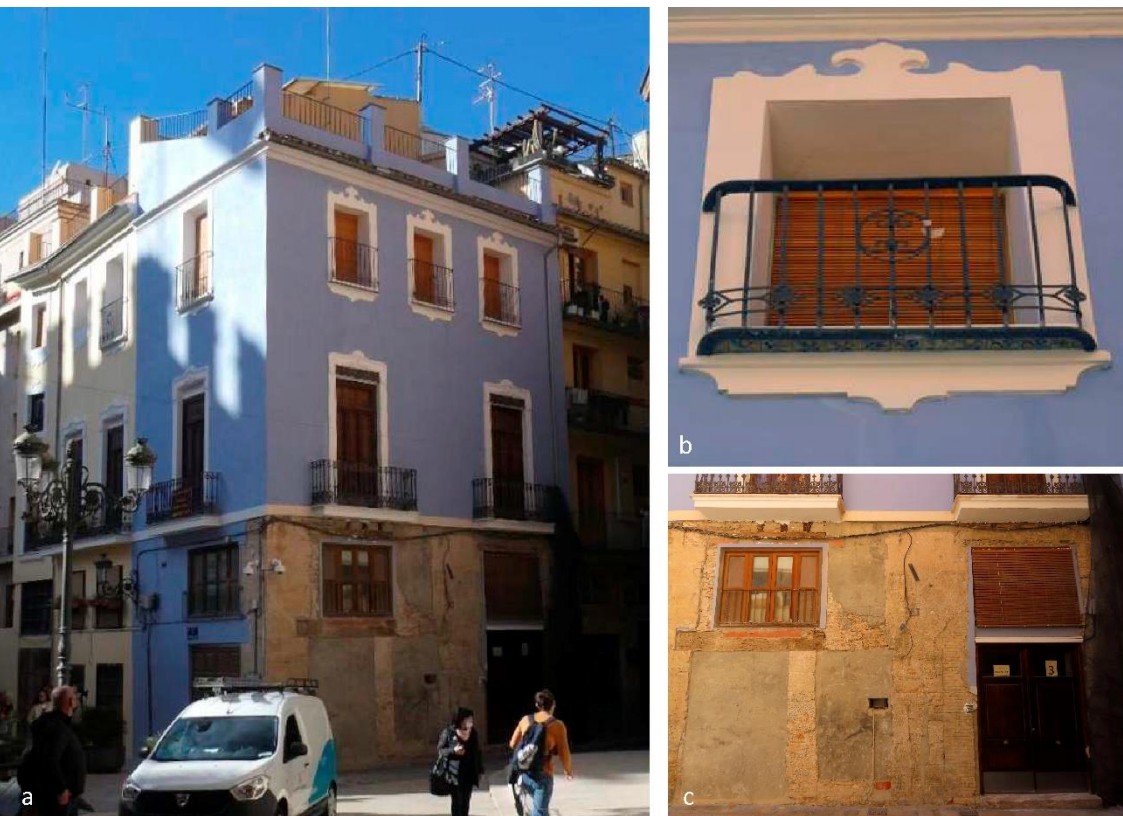

**Figure 10.** (**a**) The building at Compañía Square n. 3 on the corner of Cordellats Street; (**b**) detail of the balcony on the second floor; (**c**) detail of the ground floor. (Source: authors)

This building is known to have been the residence of doctor and writer Jaume Roig (1407 or 1409–1478). Could this be the same building where the famous author of *L'Espill* lived? It is, in fact, highly probable that at least the walls of the building are mostly those of the dwelling of Jaume Roig. The writer's façade would have been made up of half-body or even full-length windows with wooden railings flush with the wall, with no projections. Undoubtedly, the joinery was made up of blind shutters with no glazing, and in winter wooden frames with oiled linen were used to let in the light while still offering protection from the wind and partly the cold. From the exterior, no evidence is seen of the ceilings of the building, where wooden beams and joists from this period and some details may have survived the passing of time.

The building on Lonja Street no. 8, on the corner of Compañía Square and Cenia Street (Figure 11a), is shown in the 1810 engraving depicting the uprising of the Valencian people against Napoleon and already appears with the same five storeys found today. In fact, the stone doorway with rounded edges; the joinery of the main door; the format of the balcony, projecting three palms (67.5 cm) with iron braces that clear to avoid the openings (Figure 11b); the smaller balconies with rounded corners (Figure 11d); the simple bars of the railings, produced by a blacksmith's hammer and not a cast mould (Figure 11c); the full-body closed-off balconies with a square section and curved corners to avoid the use of the spiked joint, reserved only for the upper edge; and the ceramic tile of the underside of the

balconies suggests it dates to 1770–1780 (Figure 11b). The wooden eaves that look both onto Lonja Street and Cenia Street are indicative of a construction which is also characteristic of the 18th century or earlier, as in the 19th century pressure from the Academia de Bellas Artes and local regulations led to the elimination of this type of wooden eave and to the construction of finishing cornices. Under the yellow paint, recently applied, the extensive, slightly irregular and bulging gypsum rendering characteristic of the 18th century is still found, clearly visible.

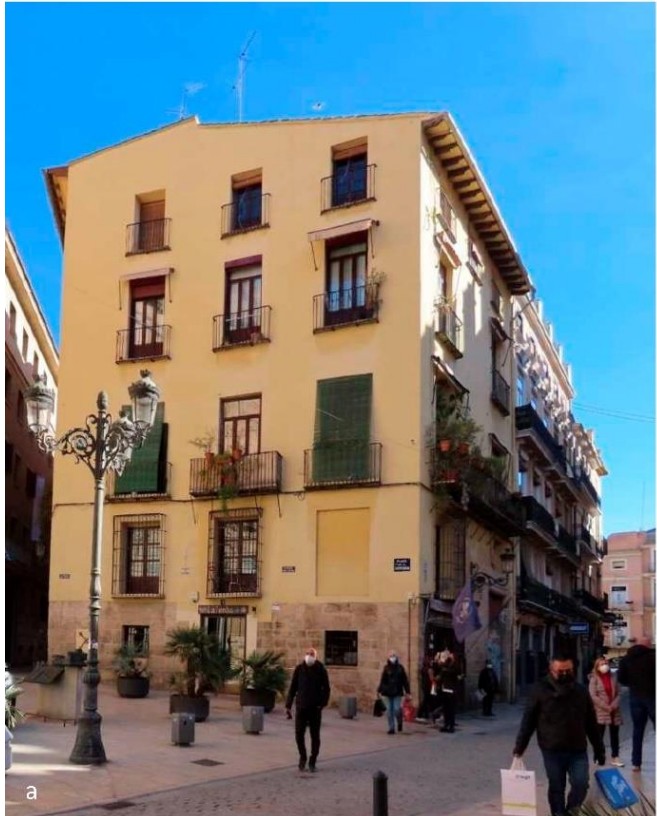
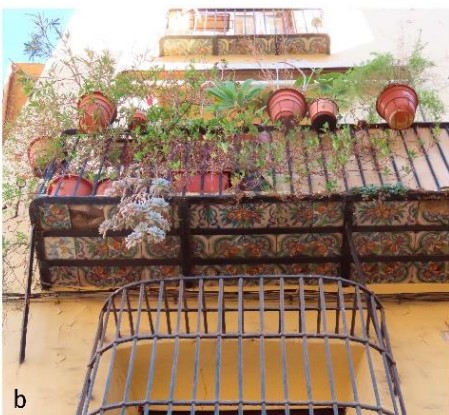
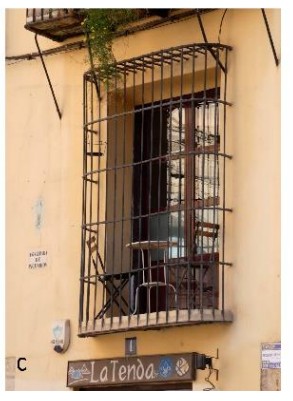
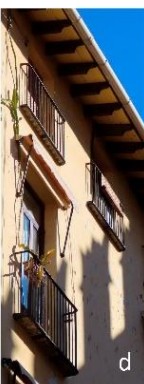

**Figure 11.** (**a**) The building on Lonja Street no. 8, on the corner of Compañía Square and Cenia Street; (**b**) detail of the balcony on the second floor; (**c**) detail of the first gate; (**d**) details of the balconies of the upper floors. (Source: authors).

The balconies on the façade of Cenia Street, project three palms (67.5 cm), feature bars, a lower filigree band and a solid ledge measuring a palm and a half (33.75 cm) on the underside of the balcony, suggesting that this façade was remodelled circa 1840. The joinery throughout the building is wooden, with shutters at least on the main floor. When the balconies were added the late 18th century joinery would almost certainly have been large blind wooden shutters where, instead of glass, frames with oiled linen would let in the light in and offer protection from the cold. The use of glass in residential buildings in Valencia took off only after 1840.

However, the stone ashlar corners have a smooth edge and no decoration; a change in wall thickness is detected between the first three floors and the last two; there is a larger separation of the mezzanine between the third and fourth floors; and the ashlar bond on the plinth, which was rougher, is also different from that used in the doorway, which was larger, thinner and only partly connected to the rest of the plinth. This suggests that there may have been an original building with ground, first and second floors which, later, in 1770–1780, was remodelled, adding the large ashlar doorway and two upper floors. All the openings were remodelled, introducing new balconies and closed balconies, a type of intervention that was extremely popular in the city at the time. All these details

point to the original existence of a medieval building, probably accessed via a Roman arch where the current doorway now stands; the walls, and perhaps the interior ceiling, were swallowed up by the 18th century intervention, giving the building the appearance it still conserves today.

In conclusion, both buildings appear to have witnessed not only the life of one of the greatest literary figures of the Valencian language and the declaration of war from the Valencian people against Napoleon, but also the construction of the Lonja: Jaume Roig's home, with fewer changes in volume, and the building on Lonja Street considerably heightened by two new floors. Both buildings are therefore several centuries old and are as important as the Lonja both in terms of the context provided and as examples of the evolution of the built material culture of housing within the city. Numerous medieval buildings like these are still conserved around the Lonja, masked to varying degrees by the transformations they have suffered through time.

### 2.7. The Urbanisation and Buildings of the Market in the 19th and 20th Centuries

This chapter examines San Fernando Street (Figure 12) and the porticoed market, the result of the confiscation and demolition of the two convents, and the central market, which replaced and expanded the porticoed market.

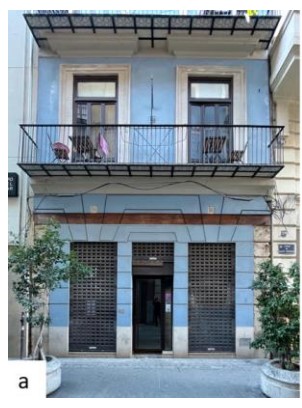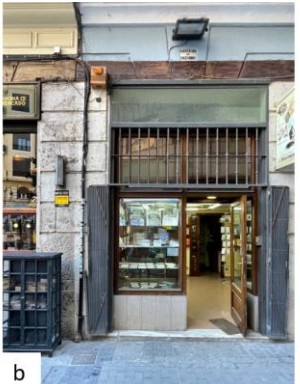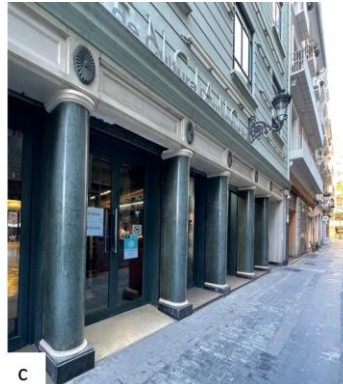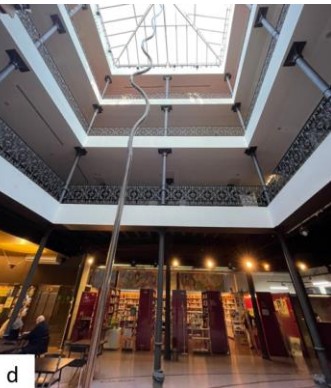

**Figure 12.** (**a**,**b**) Buildings with shops on their respective ground floors at San Fernando Street; (**c**,**d**) building called "El Siglo Valenciano" nowadays ((**c**) exterior; (**d**) interior)). (Source: authors).

### 2.7.1. San Fernando Street

San Fernando Street was established between the 1820s and 1830s and became a highly successful commercial gallery during the 19th century. The buildings on either side of the street were designed in a homogeneous neoclassical style (Figure 12a,b), with rustication on the ground floor which even spread to the wooden lintels, and three upper floors, the first two of which had continuous balconies projecting three palms with simple bars and struts, as well as dressed balcony windows.

This uniform urban design project was similar to another one dating from the late 18th century, aiming to transform Miguelete Street into a corridor of Neoclassical buildings by the architect Cristóbal Sales (1763–1833), although it was never completed. It is also similar to the projects completed several years later in the nearby Redonda Square (1839) by the architect Salvador Escrig Melchor on the former slaughterhouse; and on Moro Zeit Street, Rey Don Jaime Street and La Conquista Street (1843) on the plot resulting from the expropriation and demolition of the convent of La Puridad by Antonino Sancho Arango (1805–1876) ([27], pp. 54–55, 68, 766, 774).

In just a few decades, this uniform urban layout was destroyed by the invasive design of shop windows, and worse still, the replacement of some of these buildings with different constructions, some very tall. It is currently hard to imagine the urban uniformity, which can only be sensed in some of the façades conserved from that period but interrupted by later buildings.

A notable example among the buildings on this street was "El Siglo Valenciano" (Figure 12c,d), a shop founded in 1879. This was a pioneering textile department store in Valencia, run by Bernardo Gómez. Another unique building from this period was the Hostal del Gamell, lauded in the book on spectacles in Valencia written by Hispanicist Henri Mérimée (1878–1926). He was a first cousin once removed of Prosper Mérimée ([37], p. 39), who, as already said, had also visited the market on his trips to Spain [38].

2.7.2. The New Market or Porticoed Market

A Royal Order of 5 June 1838 awarded the Ayuntamiento de Valencia the former convent of Santa María Magdalena, expropriated three years earlier through the law of Mendizábal. The space created by its demolition and the addition of nearby surroundings provided a plot for the construction of the porticoed market (1839) by the architect Franco Calatayud Guzmán (1795–1854) (Figure 13). Its U-shaped porticoed structure with a trapezoidal annexe, and the porticoes looking onto the market square, earned it the nickname porticoed market. However, as this building could not accommodate all the activity, the market square continued to be filled with awnings or *envelats* covering the stalls that lined the perimeter, as well as the adjoining streets in the *encants* or flea markets, which filled Vieja de la Paja Street and the outer sides of the market. The square, traditionally in the open air with improvised fabric-covered stalls, or at most sheltered under the building porticoes, was partially transformed into a market with a permanent structure and covered stalls. The porticoed market had a surface area of approximately 5000 m² and 200 m² of this was gained by taking that space from the market square and blocking the opening of Blanes Street onto the square. José Campo, son of the market spice trader Gabriel Campo and future Marqués de Campo, witnessed the birth of this market from the family home, located directly opposite. Once the construction of the porticoed market was completed, work began on the cobbles of the market square ([6], p. 110).

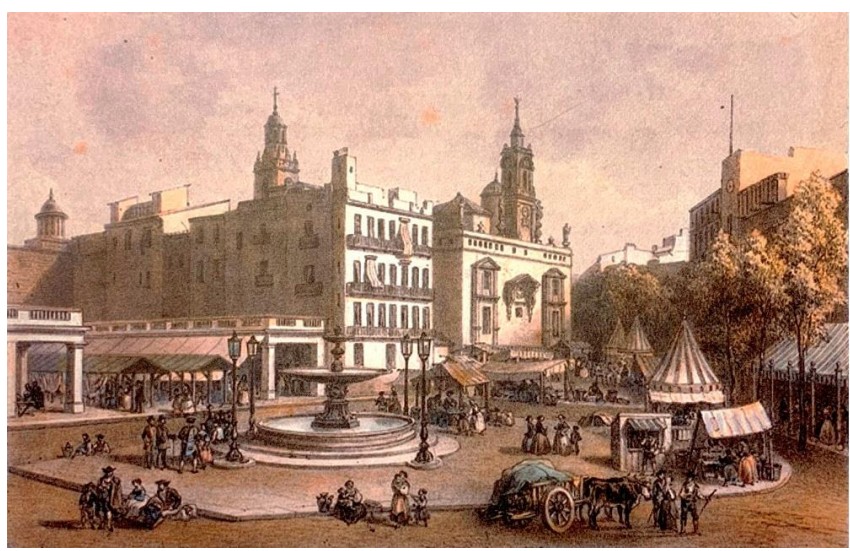

**Figure 13.** The porticoed market (1839) by the architect Franco Calatayud (Author: Deroy, 1860, [21]).

Vicente Blasco Ibáñez (1867–1928), who was born near the church of Santos Juanes, was a constant witness to the activity and colourful hustle and bustle of this market, as described in his novel *Arroz y Tartana* (1894) [45].

"Beyond, above the muddle of awnings, the zinc roof of the flower market; to the right, the two entrances to the porticoes of the new market, with the low columns painted in bright yellow (...). The two servants found their baskets increasingly heavy, and trailed after their mistress through the compact and restless crowd gathering at the entrance to the new market, whose porticoes, in the brightest afternoon sun, were as gloomy and damp

as a cave opening (...). It was there that the monotonous lull of the market was most insufferable. The ceiling of the porticoes echoed and amplified the voices of the buyers."

And the fabulous activity of the market on Christmas Eve:

"Good God...! All these people! The whole of Valencia was there. The same thing happened every year on the day of Christmas Eve. That extraordinary market, which remained open until late into the night, became a noisy festivity, the explosion of joy and din of people who among stacks of food and inhaling the scent of the thousands of things which satisfy human voracity, were rejoicing at the thought of the large amount of food to be eaten the following day. In that long square, slightly arched with narrow ends, like a bloated intestine, there were piles of food for days spreading like a rain of nourishment on tables, satisfying the great gluttony of Christmas, a gastronomic feast, which is like the stomach of the year."

At the same time, similar operations for expropriation, demolition and construction were used to establish the Market of La Boquería in Barcelona (1840), which was covered in 1914 by a large iron roof structure, and the Mercado de Abastos or wholesale food market in Cádiz (1838). Both these markets are still in full operation. However, the Ayuntamiento de Valencia soon became aware of the need for a larger—and most importantly, covered—market. The porticoed market continued in operation until it was demolished in 1916 and replaced with the new central market.

2.7.3. Central Market

From 1881, the Ayuntamiento made several attempts to construct a covered central market. In 1883, a competition was called for new designs for the new covered central market, although the winning proposal, by the architect Adolfo Morales de los Ríos (1868–1928) was never completed. In 1907, the architect Joaquín Almarza submitted a design for a prefabricated iron market, although it was rejected as it was considered too small ([23], pp. 185–186).

Finally, a competition for projects was called in 1910, with six participants applying. These included the team made up of Catalan architects Francisco Guardia Vidal (1880–1940) and Alejandro Soler March (1873–1949), both students of the architect Lluís Domènech i Montaner (1849–1923) (who also happened to be Guardia Vidal's father-in-law). Other participants were the pair of Catalan architects Lluís Homs i Moncusí (1868–1956) and Josep Pujol i Brull (1871–1936); Vicente Rodríguez Martín (1875–1933), Valencian architect and one of the main creators of the Valencian Exposition, which was taking place at the time; and Francisco Mora Berenguer (1875–1961), Valencia architect and author of the Palacio Municipal for the 1909 Valencian Exposition.

Francisco Mora probably had a particular interest in this project. In 1907, he had been commissioned to design Casa Ordeig at the market square 13, to replace a former workshop dwelling in the market, the birthplace of Valencian politician Juan Navarro Reverter (1844–1924), Minister for Tax in Madrid and yet another famous person born near the market. The first project by Mora, taking inspiration from the Art Nouveau medievalism of Josep Puig i Cadafalch (1867–1956), was modified by Mora in 1908, reaching its definitive form, a neogothic Art Nouveau similar to that of Lluís Domènech i Montaner [9].

Following a bitter controversy that was not fully resolved, the winning proposal was that of the architects Francisco Guardia Vidal and Alejandro Soler March. Of the rest of the projects presented, only the proposal by Francisco Mora is still conserved. This project clearly shows the plan for the Colón market, designed and built years later in the Ensanche district by Mora, also taking inspiration from the architecture of Domènech i Montaner, but with three naves instead of one. Mora also designed two standalone buildings: a circular flower market reminiscent of Catalonian and Austrian Secession architecture; and a kiosk for selling the typical and local tiger nut milk, with a triangular floor plan with the corners cut off, taking inspiration from similar examples such as Olivella kiosk and Bar Torino at the Valencian Exposition, which was still taking place [10].

The project by Francisco Guardia Vidal and Alejandro Soler March, which was eventually built, was also greatly inspired by the Modernist architecture of their teacher, Lluís Domènech i Montaner. Here, a highly irregular plot is given a polygonal geometry, which produces a striking building, finished off by two brick pavilions to be used as offices. This project for the new market entailed the demolition of two elongated blocks of the market square, adjoining the church of Santos Juanes, and a third smaller one beside what is currently Palafox Street. The construction of the central market, which continued until 1928, encountered several problems along the way, starting in 1919 when the architects who had designed the project resigned from the site management and were replaced by the Valencian architects Enrique Viedma Vidal (1889–1959) and Ángel Romaní Verdeguer (1892–1973). The central market that was finally built (Figure 14) absorbed even more surface area from the market square, which effectively became a street with small expansions. Of its total surface of 8160 m$^2$ the main body accounts for 6760 m$^2$ and the fish market for 1400 m$^2$.

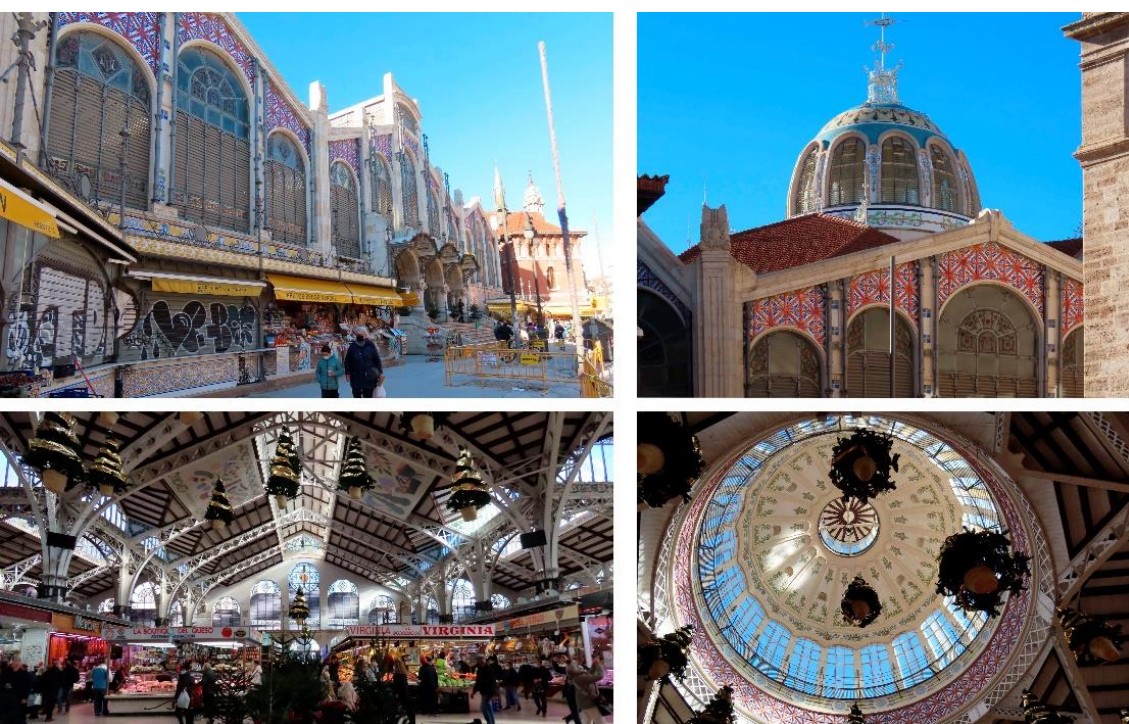

**Figure 14.** The Central Market today (source: authors).

### 2.8. The Dwellings

The area around the Lonja, as well as including these major buildings and the public space in the square, is strongly characterised by tall residential buildings with narrow plots, conferring a unique appearance to this part of the city. As mentioned earlier, the surroundings of a building as notable as that of the Lonja are completely inextricable from the monument itself, to the point that its value depends on what is found around it. The buildings of the market with its workshops or displays on the ground floor have been and continue to be one of the most important elements of the setting.

As seen earlier, little attention is paid to detail in these buildings in the first plan of Valencia, drawn by Mancelli in 1608. Mancelli draws all these buildings practically identical, with a gable roof, an entrance door on the ground floor and two windows on the first floor. Although the drawings of the buildings are extremely simple, Mancelli also provides further information: these are narrow plots, with equally narrow façades looking onto the street. The buildings are grouped into blocks where some plots are deeper. As indicated previously, the market area was close to the Moorish wall so the surroundings of the Lonja, as it stands at present, are the result of two different processes. On the one hand,

in the area within the city walls, the existing fabric of houses with courtyards and narrow streets is transformed, gradually replacing or transforming the buildings (Figure 15a), while on the other, in the area outside the walls, the territory is colonised by medieval *pueblas* or settlements with parallel homogeneous plots which run perpendicular to irrigation ditches or paths [27] (Figure 15b). In both cases, these "Christian houses" consisted of two floors (ground and first floor), with a space on the ground floor located by the entrance from the street and designed for use as a shop or artisan workshop, a space at the back towards the interior courtyard, used for the kitchen and alcoves, and a room or chamber at the top for use as a bedroom ([17], pp. 86–87) (Figure 15c).

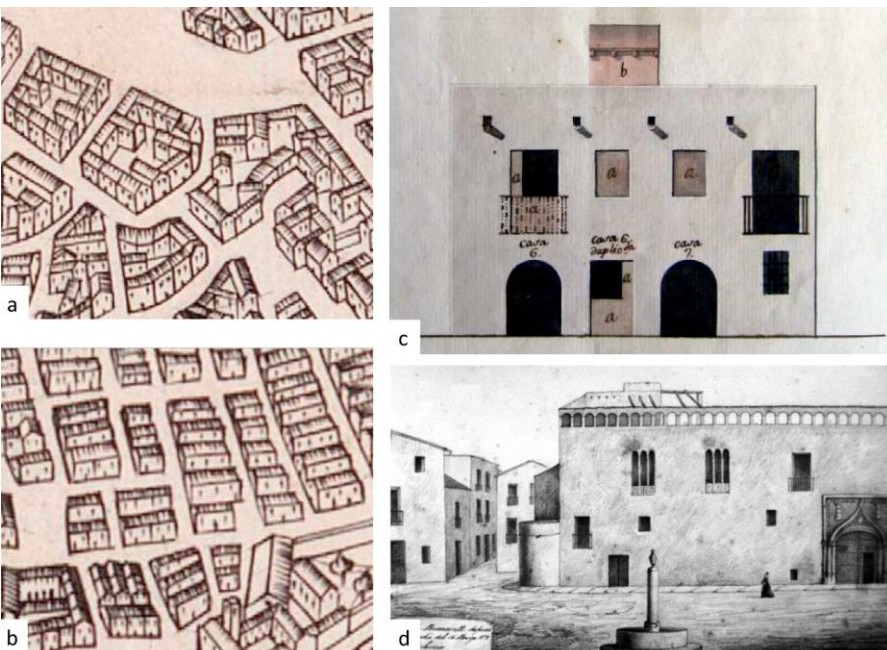

**Figure 15.** (**a**) Detail of the houses as a result of the transformation of the Islamic city (plan by Mancelli, 1608, [14]); (**b**) detail of the houses resulting from the construction of the new medieval city (plan by Mancelli, 1608, [14]); (**c**) plan of a house of the time (AHMV).; (**d**) image of the Lonja and the houses that surrounded it [21].

In addition, throughout the 14th century, the Consell worked expressly on straightening and widening streets, creating squares and new streets, closing the cul-de-sacs to prevent waste from accumulating in them [22] or opening them up to create new connections [22]. An interesting example is the order of 1383 to rectify the outline of the streets beyond the Gate de la Boatella ([22], p. 1526). The streets were also organised by eliminating any shutters, benches, protrusions, etc., which invaded them, hindering circulation [22]. As mentioned earlier, in 1447 a fire destroyed some of the houses in this part of the city so the planned reconstruction included blocks with parallel plots open to perpendicular streets.

Therefore, Mancelli's depiction of this part of the city in 1608 is undoubtedly the result of all these operations and although the plan appears limited in detail, the complex blocks stemming from the Moorish structure, the parallel plots resulting from the *pueblas*, and the blocks affected by the fire and reconstructed with blocks between parallel streets can be seen clearly.

On the plan drawn by Father Tosca in 1704, almost a century after Mancelli's plan, the buildings in the surroundings of the Lonja appear greatly transformed. It shows the porticoes on several sides of the market square in buildings with a ground floor and an additional four storeys, constructions that were quite tall for their time. These porticoes (Figure 16c,d) are the ones recorded by the numerous travellers who immortalised the market square in the first half of the 19th century (including: Laborde 1811; Trichon 1834–1835; Chapui 1844) [20]. Evidence of these porticoes is found in the files from the

late 18th century in the Municipal Historical Archive of Valencia (Figure 16a,b). In some of these porticoed buildings, applications were made for permits to add balconies while in others the addition of a single storey, changing from a ground floor plus two to a ground floor plus three, was requested [27]. Archive files connected with the market square in the late 18th century include a large number of applications to build balconies or replace old balconies with newer ones (in keeping with the transformations observed throughout the city); demolishing protrusions; reconstructing the exits to the roof and adding railings; demolishing and building new houses with balconies; some inspections for houses at risk of ruin. In 1783, there was an inspection of the portico of a house in the convent of La Merced (AHMV 1783). Different files from 1796, and especially 1797, refer to the repair of damage following the fire which had taken place in the market: "repairing the exit to the roof, repairing windows and other works required following the market fire", "rebuilding a house, remodelling the windows and balconies of the attic, following a fire", "rebuilding a charred apartment and incorporating a balcony", "leaving the house as it was originally", etc.

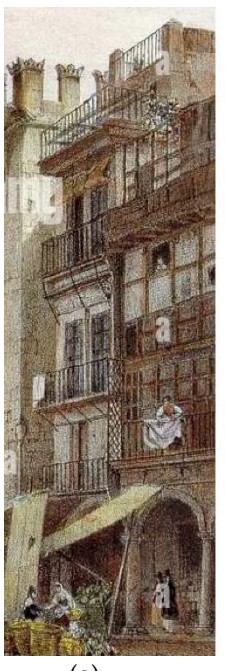 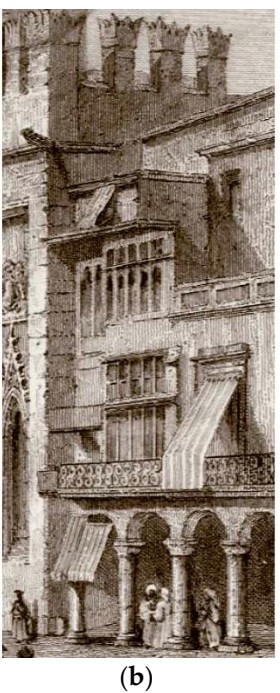 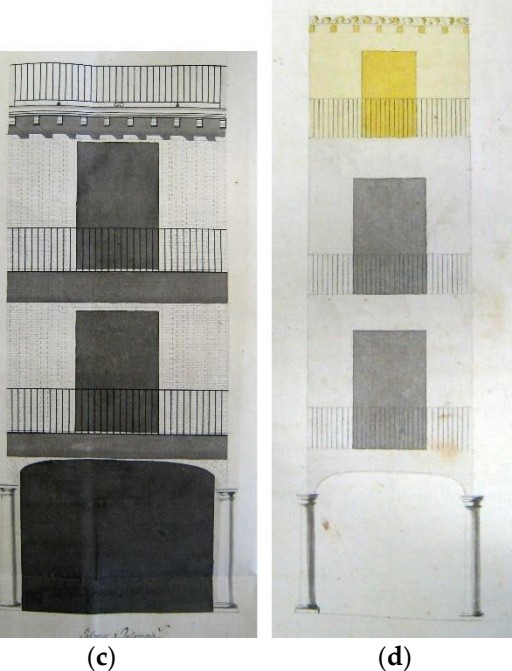

(**a**)  (**b**)  (**c**)  (**d**)

**Figure 16.** (**a**,**b**) Two images of the buildings with porticos facing the plaza (from the engravings, respectively, of Chapui, 1844, y Rouargue, 1850, [21]); (**c**,**d**) two drawings from 1797 from the Historic Archive of Valencia that correspond to reform projects for buildings with a portico on the ground floor (AHMV).

In the surrounding streets, Tosca also depicts buildings that are unusually tall for that period, with no porticoes, but with wide openings for workshops or shop windows. All through the 18th century, AHMV files frequently requested permission to work on the doors of the ground floor. There are many files related to the opening of doors, "making a door jamb" or "making two door jambs", in some cases with "ashlar stone", so it is understood that this is about reconstructing one or two door jambs on the ground floor, perhaps to widen them.

There is currently no trace of the porticoes of the market square that disappeared in the second half of the 19th century, as described by Teodoro Llorente in 1889: "the porticoes of the houses disappeared, their narrow windows were widened, the small wooden balconies were replaced with other iron ones, conferring a modern appearance to the entire building". Also, around this time, Blasco Ibáñez described the buildings around the market square: "groups of narrow façades, clustered balconies, walls with lettering, and on all the ground

floors, shops selling foodstuffs, clothing, drugs and drink, with doors displaying the names of the establishments, as many saints as are found in heaven and as many common animals of all types are found" [45].

Furthermore, the plans and views from the 19th century show a row of buildings opposite the Lonja and beside the church of Santos Juanes, even seen in photographs from the early 20th century. These buildings were demolished to make space for the construction of the central market. On the corner of this block (identified as number 396 on the plan from 1831) close to the church of Santos Juanes, was the building of the Principal, described by Blasco Ibáñez as "a very poor building, a miserable guardhouse, whose door is guarded by the watchman, weapon in hand, with a bored demeanour, and a bayonet grazing the off-duty soldiers, who are eating their tasteless meal while contemplating the sea of foods which spreads over the square" [45].

In these photographs from the early 20th century, the top end of the same block, looking onto Conejos Street and the porticoed market was made up of a series of narrow houses of different types commonly found in the area of the Lonja (Figure 17): the workshop dwellings, with a workshop and/or shop on the ground floor leading to one- or two-storey dwellings and the chamber or attic store which form part of the medieval legacy of the city (Figure 17a); the stairway houses are small residential buildings, mostly built during the 19th century, combining a workshop or shop on the ground floor with a small entrance to a narrow stair for accessing the independent dwellings on the upper floors, usually one per floor, with minimal dimensions (Figure 17b); the workshop dwelling with a double shop window on the ground floor, possibly the result of merging two workshop dwelling plots (Figure 17c). These can be considered a derivation of the single workshop dwelling given that it functioned in the same way, with access to the upper floors through the shop on the ground floor. In some of the areas surrounding the Lonja it is also possible to find communal residential buildings with two, three or four dwellings per floor with access to the common staircase through a door on the ground floor, usually connected to ground floor commercial premises with shop windows (Figure 17d).

There are currently relatively few examples of medieval workshop dwellings, but some can still be found at different points within the market square and Collado Square and Ercilla Street, Palafox Street, etc. Some of these houses were uninhabited on the upper floors as the living space was accessed through the store. In one of these, a ceiling was identified, with wooden beams, laths and ceramic tile, and given the similarities of its decoration to one of the ceilings of the palace of the Valeriola family, it could date to the 15th century [27]. In some cases, interventions were carried out to establish direct access to the ground floor staircase from the street, separating the living and commercial spaces. Workshop dwellings with two shop windows on the ground floor were extremely common throughout the southeast side of the market square, which featured a series of buildings of this type. Many of these still survive, and both the ground floor shops and the upstairs dwellings have been remodelled.

The surviving buildings, which have undergone numerous interventions and transformations, have become a hybrid mix of architectural styles and elements. Many of these transformations are documented in the files of the AHMV [27] and their history can be briefly examined for a more in-depth understanding of the value of these survivors of history. In the 18th century, closed-off iron balconies were inserted, the loggias or small chamber windows were blocked off, full-length windows were opened onto the balcony, and one or two floors were also added. The widespread presence of balconies in modern Valencia is explained by an 18th century transformation changing a city with plain façades with windows into a city with balconies on all buildings and heights. Closed-off balconies with wrought iron plates and bars, ceramic flooring and struts to support the structure are from this period. In the mid-18th century, the size of ceramic floor tiles changed (from 11.25 cm per side to 22.5 cm per side, or from half to one Valencian palm), lightening the load of the "cages" as fewer plates were needed to secure the tiles. The installation of balconies was associated with opening up windows to create doors for accessing the

balconies, in most cases adding joinery, and shutters with no glass, usually folding into the interior side of a thick façade wall seen on the edge of the windows. Interventions like these are widely seen in the buildings around the Lonja, although the examples currently found on the stretch from the market square between the Lonja and Tabla de Cambios Street and between Encolom Street and Bolsería Street are particularly interesting. On these stretches, we find two groups with three and four dwellings which perfectly illustrate these interventions, as well as others from the 19th century (Figure 18).

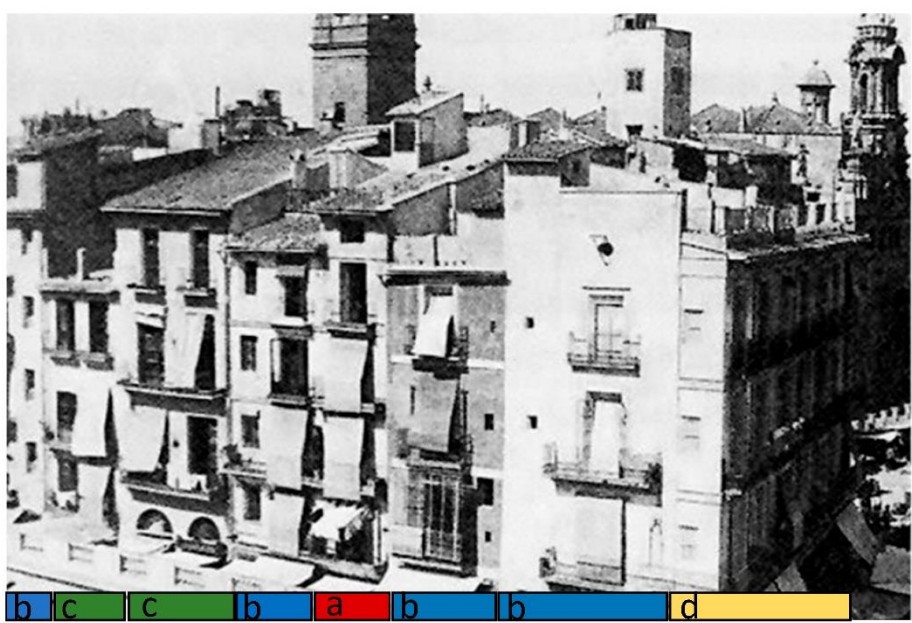

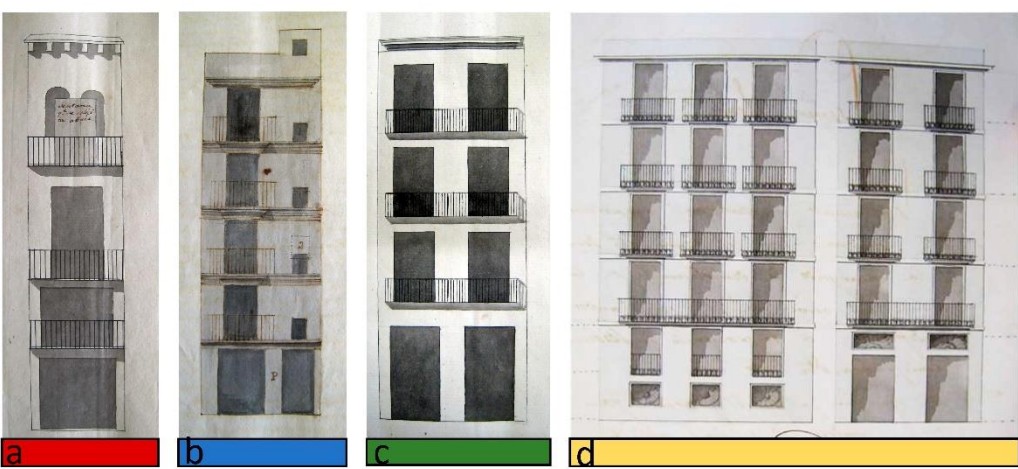

**Figure 17.** In the upper part, a photograph from the beginning of the 20th century where the different types of houses in the market area can be recognised; in the lower part, four drawings from the Historic Archive of Valencia that illustrate the different types of houses: (**a**) workshop dwellings (1793); (**b**) stairway houses (1794); (**c**) workshop dwelling with a double shop window (1796); (**d**) communal residential buildings (1854).

In the first half of the 19th century, there continued to be intensive updating of the façades of pre-existing buildings [27]: closed-off balconies were still used, although they were progressively supported by brick brackets which eliminated the need for struts; iron balustrades were added on roofs; in some cases, the access to the living quarters of the workshop dwellings was separate from the ground floor shop thanks to a door added on the façade. The access to the dwelling or dwellings in the upper part of these buildings has always been a weak spot, causing abandonment and deterioration. It is still

possible to find abandoned buildings of these characteristics, although possible compatible solutions have been put in place, very similar to those already proposed in the 19th century. In this regard, in Collado Square (Figure 19), the houses on the north side have mostly undergone interventions to divide the commercial openings from the ground floor to create an independent entrance leading to the upper floor dwellings.

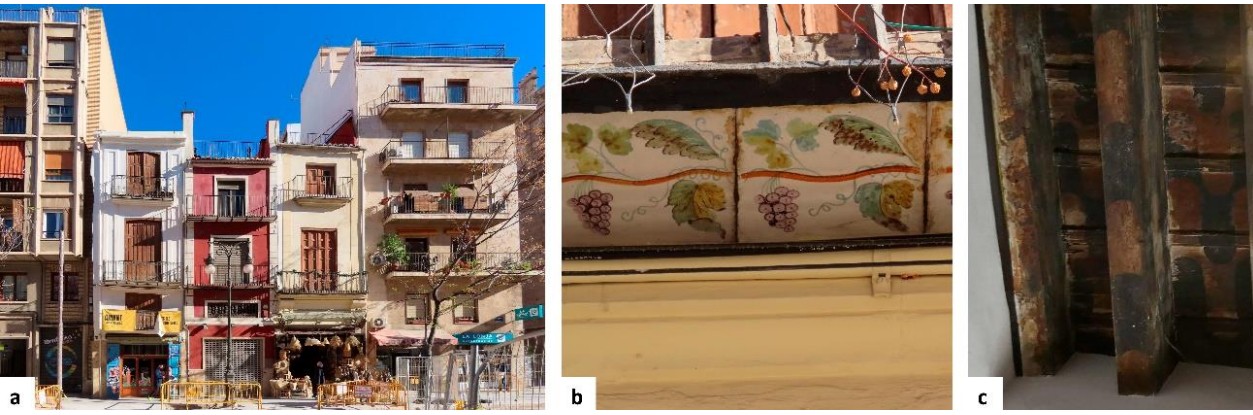

**Figure 18.** (**a**) Buildings on the stretch from the market square between the Lonja and Tabla de Cambios; (**b**) detail of a balcony from the mid-18th century; (**c**) ceiling with wooden beams, laths and ceramic tile that could be dated to the 15th century. (Source: authors).

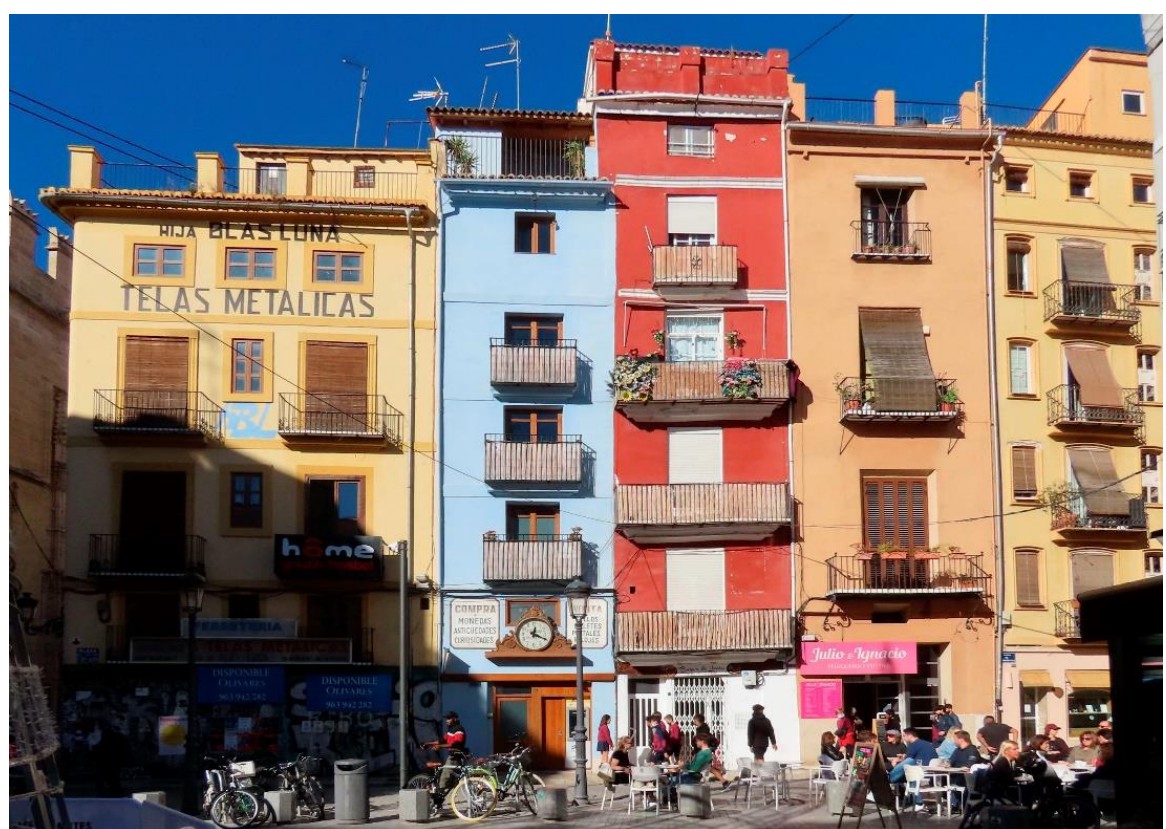

**Figure 19.** Buildings in Collado Square. (Source: authors).

In the second half of the 19th century, but especially in the final twenty-five years, windows with glass were either added outside the shutters or replaced altogether, while wrought iron bars were incorporated into balconies, increasing the potential for decoration and variety. On Palafox Street (Figure 20), it is possible to observe a row of late 19th

century workshop dwellings (except for the one on the corner with Blanes Street which is from the late 18th century). These are probably the result of remodelling interventions in previous buildings. In these it is possible to observe how decoration is added to the façade with rustication (used in Valencia from the 1830s), strips separating floors all along the façade, brackets and pilasters adding reliefs to the rendering, which becomes more prominent thanks to colour contrasts; wrought-iron balconies resting on brackets and dating to 1870–1880; and wooden joinery with glazing and *frailero* shutters (which have not been replaced in restoration interventions).

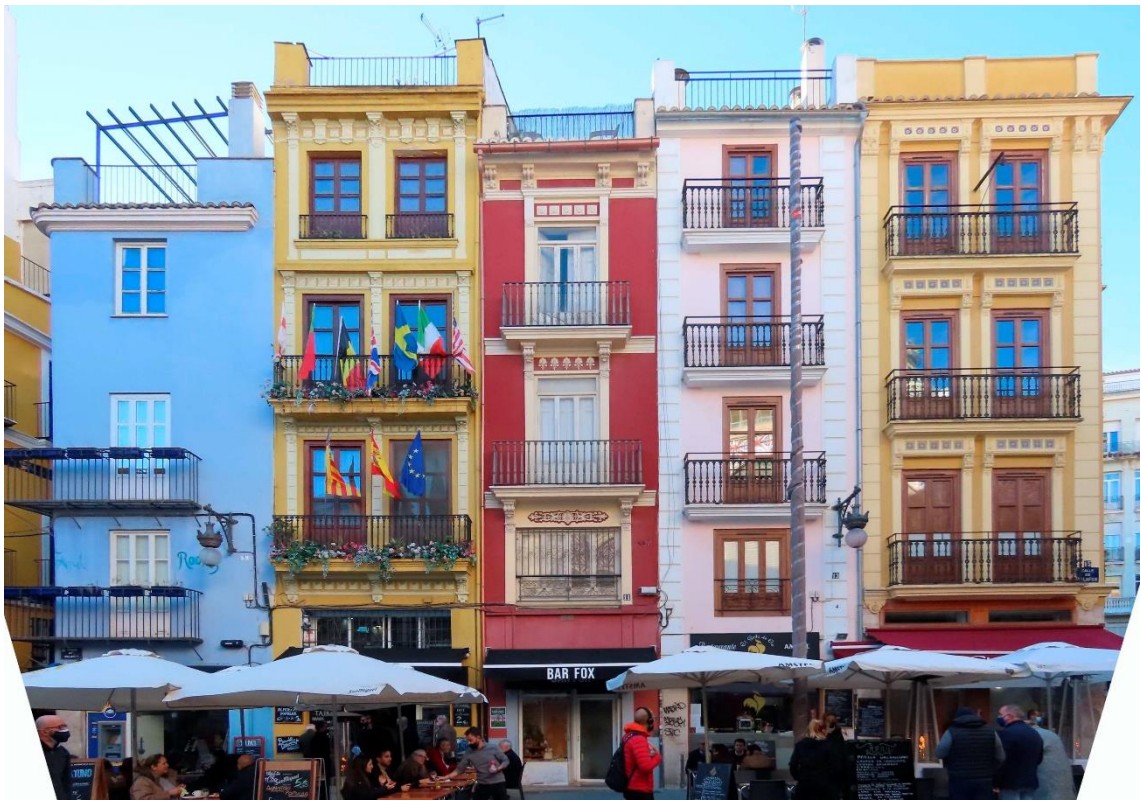

**Figure 20.** Buildings in Palafox Street. (Source: authors).

Some stairway houses remain on d'En Gall Street, around the corner from Palafox Street (Figure 20). The architectural elements of these two dwellings go back to the second half of the 18th century or early 19th century. Both buildings still conserve the staircase on the façade, visible thanks to the entrance door beside the ground floor shop window and the small windows of the stairwell. Files have been found [27] from the late 19th and early 20th century proposing to move the stairway on the façade closer to the second set of stairs, thus freeing the façade bay and providing a span for a larger room. This type of intervention is usually linked to the widening of the small staircase windows, converting them into windows or full-length balcony doors. Interventions like this can be seen on Pere Compte Street.

Immediately surrounding the Lonja, there are also examples of notable communal residential buildings, probably the result of the merging of medieval plots with narrow workshop dwellings and a remodelling process taking place from the mid-19th century onwards. These buildings include neoclassical mouldings and pediments, wrought iron balconies, and—although covered by paint—the ashlar effect decorating the entire façade as can be seen in photographs from the late 20th century. Buildings of this type, with a communal entrance from the street used to access dwellings, generally two per floor, can be found in several locations around the Lonja, from the market square to Collado Square, and also including Derechos Street or d'En Gall Street, etc.

From the second half of the 19th century, the city of Valencia was immersed in a frenzied rhythm of remodelling and renovation of both the public space and the housing stock. Squares and streets were opened up, new alignments were created in order to widen the small narrow streets of the medieval layout, and older insalubrious residential buildings disappeared to make way for collective housing blocks with improved ventilation and functionality. This process also reached the surroundings of the Lonja. If the "Proyecto de Reforma Interior" by L. Ferreres (1891) had been executed, at present the surroundings of the Lonja would be drastically different and there would be no call for a study of these characteristic dwellings in the market areas. The idea of opening up wide avenues and streets to replace the buildings around the Lonja and the market was a constant during almost half of the 20th century, as recorded in numerous plans and projects. Among these, it is worth noting the plan of "Nuevas líneas para la Reforma Interior de Valencia" by J Goerlich in 1929, which foresaw a complete transformation of the entire area, connecting to Ayuntamiento Square, and opening up the Oeste Avenue behind the central market. The renewal plan, as well as opening Oeste Avenue as far as the church of Santos Juanes, especially affected the southern access to the square from San Fernando Street as María Cristina Avenue and Ayuntamiento Square were now connected. The creation of the new avenue led to the demolition of different blocks of housing up to San Vicente Street and the construction of the collective housing buildings is still seen in the same spot today. These dwellings, mostly built in the 1930s, were on a far greater scale than the construction in the square (although in keeping with porticoed buildings which previously must have been in this area), as well as curved chamfers in stark contrast with the smooth linear façades of this part of the city (Figure 21).

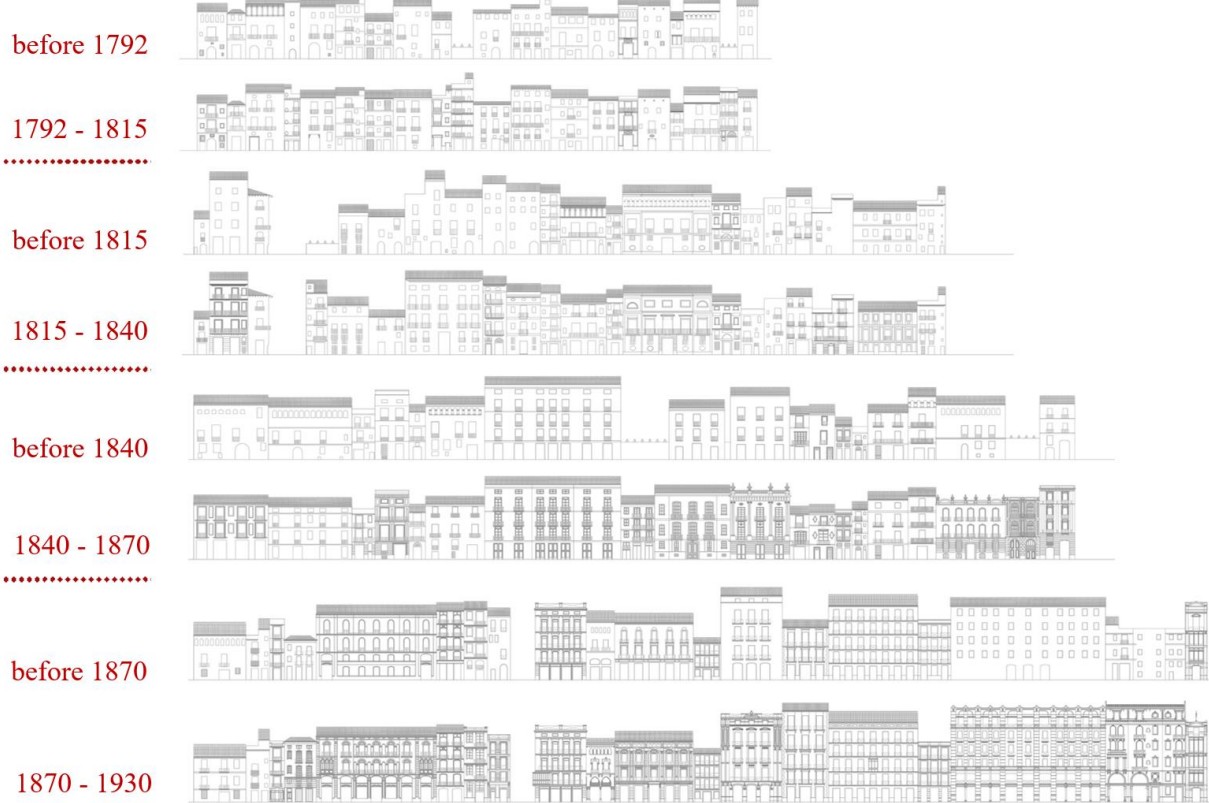

before 1792

1792 - 1815

before 1815

1815 - 1840

before 1840

1840 - 1870

before 1870

1870 - 1930

**Figure 21.** The transformation of the image of the city through the conversion of the facades, demolitions and new buildings during the 1792–1930 period. The reconstruction has been completed through real projects from the files of the Archivo Histórico Municipal de Velencia [27].

## 3. Conclusions

When the Lonja de la Seda was declared World Heritage by UNESCO in 1996, the building was viewed as an object independent from its surroundings. Between 1996 and 2016, when a review of the documentation for this declaration was carried out, no modifications were made to the perimeter of the declaration, which continued to be linked solely to the building. This article has aimed to prove that the Lonja de la Seda is not an isolated object detached from its surroundings. In addition, its urban setting remains closely connected—in urban, historical, architectural and ethnological terms—to the actual building of the Lonja. The surrounding buildings, whether non-residential (markets and churches) or residential (public spaces such as streets and squares), are all part of an indivisible complex.

The Lonja is part of the enclave of the market of Valencia, which is a thousand years old and has evolved through a history of configuration, transformation and adaptation, which continues even today. This setting is made up not only of monumental buildings like the church of Santos Juanes, the Jesuitical church or the Central Market, but also—and above all—of a residential fabric made up principally of dwellings and ground floor shops woven into it. Furthermore, in most cases, these dwellings, which are several centuries old or even date to the Middle Ages, show a tendency that has favoured the repair, modification and transformation of the building rather than the complete demolition and construction of a new building, so preserving evidence of all the centuries of its history to the present day.

This study has attempted to merge historical, documentary, urban, architectural and material perspectives, employing indirect and direct methods in order to prove that the surroundings of the Lonja are as deserving of attention and protection as the Lonja building itself. The unique architectural and constructive nature that earned the building its designation would not have been possible without its surroundings, linked to commercial activities for years. In this respect, while the Plan de Protección de la ciudad de Valencia (2020) considered these circumstances when establishing an area of protection around the monument (see Figure 4), it is striking that the UNESCO designation for the Lonja does not include a buffer zone, as seen in numerous other cases. Following the research detailed in this text, a proposal is put forward for this perimeter to at least match the protection perimeter established by the Ayuntamiento de Valencia, although this buffer zone could in fact be expanded to cover the entire area associated with the market's activities over the centuries, the artisanal and manufacturing area, and the homes of the merchants and artisans who sold their products from their workshops in the market's area of influence (Figure 22). Finally, it should be noted that the objective set out in this text was the provision of elements for the definition of tangible values (buildings, urban structure, materials and constructive techniques, typologies, etc.), as well as intangible ones (history, transformations, uses, functions, experiences, etc.), for a specific setting within the historic city of Valencia. However, the multi-faceted methodology used in this research could be extended to other settings within the same city, as well as to other historic cities.

The specific case of the Lonja de la Seda of Valencia, declared a World Heritage Site for its exceptional value regardless of its surroundings and the city that has actually produced it, is not an isolated case. Many other outstanding buildings have been declared autonomously and independently of their surroundings and the proposed case aims to stimulate a discussion towards a possible revision of the declarations that does not contemplate the relationship with their respective environments, so that they can be expanded and therefore include the environment not only as a buffer zone but as part of the same declaration. The case of La Lonja shows us how the essence of the city lies in its dynamism. Some things change for the better, others for the worse, and others never change. The work of the guardians of architectural heritage is to ensure that things change for the better, respecting the essence of the city, which is the legacy of generations past and future. Knowledge of the material history of the city through research is essential to earn respect for this built essence in order to protect, restore and enhance it.

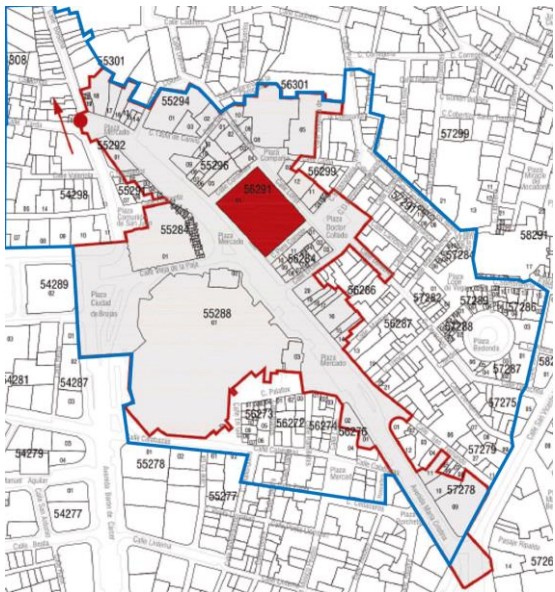

**Figure 22.** Plan of the surroundings of the Lonja de la Seda with the perimeter of the protection area outlined for the Plan de Protección de Ciutat Vella (2020) (in red) and the authors' proposal for a possible buffer zone (in blue). (Source: authors).

**Author Contributions:** Conceptualisation: C.M.; methodology, C.M. and F.V.L.-M.; documental research: C.M. and F.V.L.-M.; on site research: C.M. and F.V.L.-M.; photos: C.M. and F.V.L.-M.; writing—original draft preparation: C.M. and F.V.L.-M.; writing—review and editing, C.M. and F.V.L.-M. All authors have read and agreed to the published version of the manuscript.

**Funding:** This research was self-funded by the authors.

**Institutional Review Board Statement:** Not applicable.

**Informed Consent Statement:** Not applicable.

**Data Availability Statement:** Not applicable.

**Conflicts of Interest:** The authors declare no conflict of interest.

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
