# Peer review of "The Protection of the Historic City: The Case of the Surroundings of the Lonja de la Seda in Valencia (Spain), UNESCO World Heritage"

_2673-8945, doi:10.3390/architecture3040033_

Round 1

Reviewer 1 Report

as in attached

Author Response

REVIEW REPORT 1

R: The abstract should be inclusive and though it has included the problem statement, the research question, and the methods. Yet, it has to add the main results and conclusions. These parts should be added to the abstract, because, the abstract must stand alone.

A: This has now been modified.

R: The adopted methodology of the research is not ideal; so, adding ‘The Research Limitations’ as an independent item (which will be described all neglecting analyses) will be useful.

A: The methodology section has now been expanded as it did not go into sufficient detail when addressing the state of the art, objectives and methodology used in the study.

R: The conclusions should be involved a comparison between the current outcomes and the other relevant studies’ outcomes to mark the similarities and differences. This can help to suggest new directions of the future researches.

A: The conclusions section has been expanded considerably, aiming to provide a clear explanation of the advances identified in this study.

Reviewer 2 Report

TO THE AUTHOR

The authors are congratulated and urged to improve the document 'Elements for the definition and protection of the historic city. The case of the surroundings of the Lonja de la Seda in Valencia (Spain), Unesco World Heritage', both in its structure and in the fulfilment of the objectives, methodology, references, and practical approach to the problem of built heritage; however, the only pertinent observations that I will be able to comment on are:

Title

The title is very large; a maximum of 16 words is recommended.

Abstract:

In general, it should be completely restructured and highlighting in a clearer way the objective of the research, the methodology, the advantages over the results and conclusions.

What is the methodology?

Introduction:

Improve the wording and structure of the paragraphs because it generates very large paragraphs that dilate the objective idea of reading.

There is very little relevant information and assumed for the investigation, it will be necessary to clarify and justify us with greater investigative criteria.

Methods

Restructure the paragraphs completely because it tells us nothing about the research method used.

It does not describe what was the method used for the selection of the research case studies, the method needs more information such as the protocols that were followed and existing variables in the study, etc.

Conclusion

The general description of the conclusions does not generate a relevant contribution.

Author Response

REVIEW REPORT 2

R: The authors are congratulated and urged to improve the document 'Elements for the definition and protection of the historic city. The case of the surroundings of the Lonja de la Seda in Valencia (Spain), Unesco World Heritage', both in its structure and in the fulfilment of the objectives, methodology, references, and practical approach to the problem of built heritage; however, the only pertinent observations that I will be able to comment on are:

R: The title is very large; a maximum of 16 words is recommended.

A: The difference between the title (10 words) and the subtitle (17 words) has been highlighted.

R: Abstract. In general, it should be completely restructured and highlighting in a clearer way the objective of the research, the methodology, the advantages over the results and conclusions.

A: The abstract has now been expanded 

R: Introduction. Improve the wording and structure of the paragraphs because it generates very large paragraphs that dilate the objective idea of reading. There is very little relevant information and assumed for the investigation, it will be necessary to clarify and justify us with greater investigative criteria. The methodology section has now been expanded as it did not go into sufficient detail when addressing the state of the art, objectives and methodology used in the study, etc.

A: The methodology section has now been expanded as it did not go into sufficient detail when addressing the state of the art, objectives and methodology used in the study

R: Conclusion. The general description of the conclusions does not generate a relevant contribution.

A: The conclusions section has been expanded considerably, aiming to provide a clear explanation of the advances identified in this study

Reviewer 3 Report

The paper is well structured and clearly developed. and worth acceptance in present form.

Yet, to improve efficacy, I'd suggest - if possbile - to insert one or two as wide as possible views of the Lonja and its sourroundings that are  thoroughly analised but would benefit from an oveall view or at least an aerial image.

Author Response

REVIEW REPORT 3

The paper is well structured and clearly developed. and worth acceptance in present form. Yet, to improve efficacy, I'd suggest - if possbile - to insert one or two as wide as possible views of the Lonja and its sourroundings that are  thoroughly analised but would benefit from an oveall view or at least an aerial image.

A: Aerial photographs, maps and historic views have been added.

Round 2

Reviewer 2 Report

The authors should be congratulated for improving the document 'ELEMENTS FOR THE DEFINITION AND PROTECTION OF THE HISTORIC CITY. The case of the surroundings of the Lonja de la Seda in Valencia (Spain), UNESCO World Heritage' both in its structure and in the recommendations mentioned above, however, the only outstanding observations that I can comment on are:

1.- The introduction is clearer and describes well the place of study.

2.- It has very large paragraphs that do not allow easy reading, for example: Paragraph 1.1. Some definitions for further exploration of 447 or paragraph 1.2. State of the art, objectives, and methodology of the research paragraphs of 332 words, etc., please try to divide them without losing the main idea of the document.

3- The surroundings of the Lonja: points for consideration, with the support of narrative review and graphic restructuring, the paragraphs have a better understanding of the research. Graphics remain a modest element in your research; try to compose infographics rich in evidence, data, and images from your research.

4.- The conclusions are described more clearly, addressing some variables and the scope of the research established by the authors; you must restructure the specific conclusions. Better describe the variables of Heritage, but do not contemplate the material description "history of the materials" of many of your examples, which would provide a greater contribution.

5.- The reference generates a modest and interesting contribution.

Author Response

REVIEW REPORT 1

R: The authors should be congratulated for improving the document 'ELEMENTS FOR THE DEFINITION AND PROTECTION OF THE HISTORIC CITY. The case of the surroundings of the Lonja de la Seda in Valencia (Spain), UNESCO World Heritage' both in its structure and in the recommendations mentioned above, however, the only outstanding observations that I can comment on are: 1.- The introduction is clearer and describes well the place of study.

A: Thank you

R: 2.- It has very large paragraphs that do not allow easy reading, for example: Paragraph 1.1. Some definitions for further exploration of 447 or paragraph 1.2. State of the art, objectives, and methodology of the research paragraphs of 332 words, etc., please try to divide them without losing the main idea of the document.

  1. The paragraphs have been divided to make the text more understandable (marked in green)

R: 3- The surroundings of the Lonja: points for consideration, with the support of narrative review and graphic restructuring, the paragraphs have a better understanding of the research. Graphics remain a modest element in your research; try to compose infographics rich in evidence, data, and images from your research.

A: Images and drawings have been added to enrich the article (marked in green): figures 8, 12, 21

4.- The conclusions are described more clearly, addressing some variables and the scope of the research established by the authors; you must restructure the specific conclusions. Better describe the variables of Heritage, but do not contemplate the material description "history of the materials" of many of your examples, which would provide a greater contribution.

A: The conclusions have been restructured and a part has been added (in green)